# Surface dynamics and history of the calving cycle of Astrolabe Glacier (Adélie Coast, Antarctica) derived from satellite imagery

Floriane Provost[1], Dimitri Zigone[1,2], Emmanuel Le Meur[3], Jean-Philippe Malet[1,2], and Clément Hibert[1,2]

[1]Ecole et Observatoire des Sciences de la Terre (EOST), CNRS UAR 830 - Université de Strasbourg, 5 rue Descartes, F-67084 Strasbourg, France
[2]Institut Terre et Environnement de Strasbourg (ITES), CNRS UMR 7063 - Université de Strasbourg, 5 rue Descartes, F-67084 Strasbourg, France
[3]Institut des Géosciences de l'Environnement (IGE), CNRS UMR 5001 - Université Grenoble Alpes, Grenoble, France

**Correspondence:** Floriane Provost - f.provost@unistra.fr

**Abstract.** The recent calving of Astrolabe Glacier in Terre Adélie (East Antarctica) in November 2021 presents an opportunity to better understand the processes leading to ice tongue fracturing. To document the fractures and rift evolution that led to the calving, we used the archive of Sentinel-2 optical images to measure the ice motion and strain rates from 2017 to 2021. The long-term evolution of the Astrolabe ice tongue is mapped with airborne and satellite imagery from 1947 to November 2021. These observations are then compared with measurements of sea-ice extent and concentration. We show that calving occur almost systematically at the onset or during the melting season. Additionally, we observe a significant change in the periodicity of sea ice surrounding Astrolabe Glacier in the last decade (2011-2021) compared to previous observations (1979-2011), which has resulted in a change in the Astrolabe calving cycle. Indeed, one can observe a decrease of the duration of sea-ice-free conditions during the austral summer after 2011 at the vicinity of the glacier, which seems to have favored the ice tongue spatial extension. However, the analysis of strain rate time series revealed that the calving of November 2021 (20 km$^2$) occurred at the onset of sea ice melting season but resulted from the glacier dislocation that took place suddenly in June 2021 in the middle of the winter. These observations indicate that while sea ice can protect and promote the spatial extension of a glacier ice tongue, its buttressing is not sufficient to inhibit rifting and ice fracturing.

## 1 Introduction

Determining the contribution of polar ice sheets to sea level rise is a major concern for the society, and a better understanding of the processes and the factors controlling ice retreat is of paramount importance for simulating the response of ice sheets to global warming (Seroussi et al., 2020; Chambers et al., 2022). Coastal glaciers in polar regions differ from mountain glaciers in temperate regions in terms of volume, catchment size and thermal state associated with complex interactions with the ocean. The presence of floating tongues with marine termini makes Antarctic glaciers more sensitive to the atmospheric and ocean dynamics (Gudmundsson et al., 2019; Olinger et al., 2019; Paolo et al., 2015; Pritchard et al., 2012; Christie et al., 2022). Monitoring of Antarctic glaciers remains heterogeneous (Baumhoer et al., 2018) and studies focus either on continental-scale

monitoring, which usually lead to commenting the evolution of Antarctica's largest glaciers (Walker et al., 2013; Rignot et al., 2019; Miles et al., 2022; Millan et al., 2022; Baumhoer et al., 2023), or on specific glaciers or groups of glaciers that receive most of the attention (Baumhoer et al., 2018). In this study, we document and analyse the evolution of Astrolabe Glacier's ice
tongue calving cycle, which has not been updated since Frezzotti and Polizzi (2002).

Astrolabe Glacier is located in Terre Adélie, (140°E, 67°S) near the French research station Dumont d'Urville. The glacier outlet is ca. 8 km wide (Figure 1a), while the drainage basin stretches as much as 200 kilometres inland. It is characterized by an ice tongue developing on the water, with a calving front 6 km wide (Figure 1a). Due to its proximity to the Dumont D'Urville research station, the glacier has been instrumented over the last few decades, focusing on the grounding zone (Drouet, 2012;
Le Meur et al., 2014). However, the last study documenting the calving cycle of the glacier's ice tongue covers the period 1940-2000 (Frezzotti and Polizzi, 2002), while recent observations show an unusual spatial extension of the ice tongue until November 2021 when a major calving event occurred (Figure 1f-i). Due to its small size, and rapid recent dynamics, Astrolabe Glacier ice tongue is not adequately monitored by global value-added products such as the NASA MEaSUREs ITS_LIVE (doi:10.5067/6II6VW8LLWJ7).

Ice calving is defined as the detachment of a smaller ice piece of ice from a larger one (Alley et al., 2023). Calving is mostly controlled by brittle processes (Alley et al., 2023) and results from the extensive opening of cracks or rifts within the ice shelf. Lateral spreading and thinning of the ice shelf can explain the formation and propagation of these fractures/rifts (Liu et al., 2015; Larour et al., 2021; Borstad et al., 2017; Alley et al., 2023). However, environmental forcing can also accelerate their propagation through hydrofracturing (Scambos et al., 2000), subglacial warm water intrusion and basal melting (Ritz et al.,
2015; Rignot et al., 2019; Pritchard et al., 2012), bending of the ice due to flexural rebound after lake drainage (Banwell et al., 2013). Examples of tsunamis contributing to open rifts and triggering calving are also reported (Liang et al., 2023; Alley et al., 2023). Another important forcing is the influence of the sea ice surrounding the ice tongues, resulting from atmospheric and oceanic dynamics (Campagne et al., 2015). In fact, changes in atmospheric and oceanic dynamics favouring the presence of sea ice can act as a protector and allow glacier expansion either by protecting the ice tongue from ocean swell and/or in the
case of landfast sea ice (i.e. sea ice fastened to the glacier/to the coastline) or by buttressing the ice tongue (Massom et al., 2001; Walker et al., 2013; Robel, 2017; Wearing et al., 2020; Massom et al., 2010; Gomez-Fell et al., 2022; Massom et al., 2018; Wille et al., 2022; Christie et al., 2022). The thickness of sea ice or ice mélange within a pre-existing rift may influence the acceleration of the rift opening (Larour et al., 2021), leading to complex calving cycles. In several cases, the disappearance of sea ice around the ice tongue front has been reported to trigger the instantaneous calving (Massom et al., 2001, 2018; Robel,
2017; Wearing et al., 2020; Gomez-Fell et al., 2022; Xie et al., 2019). However, it remains unclear whether the sea ice, and in particular the landfast sea ice, is buttressing the ice tongue preventing fracture propagation or whether it merely holds the ice tongue parts together until calving is possible. All of these processes are still poorly understood, as they exhibit strong spatial and temporal variability, which are highly difficult to document with direct observations in Antarctica.

In this study, we determine for the first time the ice tongue extension cycle of Astrolabe Glacier from an aerial photograph
taken in 1947 and high-resolution satellite imagery (ERS, MODIS, Landsat, Sentinel-2 and ASTER) covering the period 1960-2021. The archive of Sentinel-2 images is used to compute surface velocity of the ice for the entire area of Astrolabe Glacier

from 2017 to 2021. We demonstrate the added value of optical satellite imagery for monitoring fractures propagation using ice velocity and strain rate calculated from optical image correlation. We compare the frontline evolution of the ice tongue with the sea-ice extent around Astrolabe Glacier from the NSIDC (National Snow and Ice Data Center; Fetterer et al.2017). We show that calving events occur almost systematically when sea ice disappear around the ice tongue terminus, but that the rift propagation can take place in the middle of the austral winter when the ice tongue is totally embedded in sea ice suggesting that sea ice buttressing may not be sufficient at Astrolabe Glacier to prevent calving.

## 2 Data and methods

### 2.1 Satellite imagery

#### 2.1.1 Mapping of the ice front position

The ice front of Astrolabe Glacier is mapped using mainly high resolution ($< 50$ m) optical satellite imagery available in the public domain (i.e. Landsat, MODIS, ASTER and Sentinel-2). The first available satellite image was acquired by Landsat-1 on January, 29 1973 (Figure 1c). The next available acquisitions were acquired in 1989 by Landsat-4/5 and then in 1999 by Landsat-7 (Figure 1c). In 1947, the US Navy Operation Highjump took several aerial photographs of the Adélie coast, including over Astrolabe Glacier. We used a sketch derived from the photographs (https://archives-polaires.fr/idurl/1/14865) to extract the ice front position (Figure 1c). We coregister manually the photograph and attempt to compensate most of the distortions, although significant shifts remain visible. We therefore allow for an error of $\pm$ 1 km in the position of the ice front. To complement the optical dataset, we used radar imagery from ERS satellites and the Radarsat RAMP product (Jezek et al., 2013) to map the ice front position between 1996 and 1999 (Figure 1c). From 2000 to 2013, Landsat-7 and ASTER satellites provide from 1 to 3 images per year. We complete this dataset with the analysis of MODIS images (Figure 1d). From 2010, Landsat-8 and then Sentinel-2 from 2017 provide regular high-resolution coverage of ice front evolution (Figure 1e). The combination of these two satellites allowed to monitor the calving of November 2021 with daily acquisitions (Figure 1g-i). A total of 113 images are analysed and the evolution of the ice front is manually mapped. Finally, the area of the floating tongue is estimated considering an arbitrary reference grounding line position (Bindschadler et al. 2011; Figure 1a). A more precise delineation of the grounding line has been proposed by Le Meur et al. (2014).

#### 2.1.2 Ice velocity monitoring from optical images

Satellite imagery is commonly used to compute the ice velocity using image correlation techniques (Avouac et al., 2006; Leprince et al., 2007; Rignot et al., 2011; Mouginot et al., 2017; Millan et al., 2022). These techniques consist of matching pixels from one image to another in order to obtain the shift in position of a given feature over time. Several studies have shown the interest of this technique for monitoring ice surface velocity (Dehecq et al., 2015; Altena et al., 2019), especially in polar regions (Joughin et al., 2018; Millan et al., 2022). We used the GDM-OPT-ICE service (Provost et al., 2022; Stumpf et al., 2017) to compute ice displacement time series. The GDM-OPT-ICE service allows for the precise co-registration of

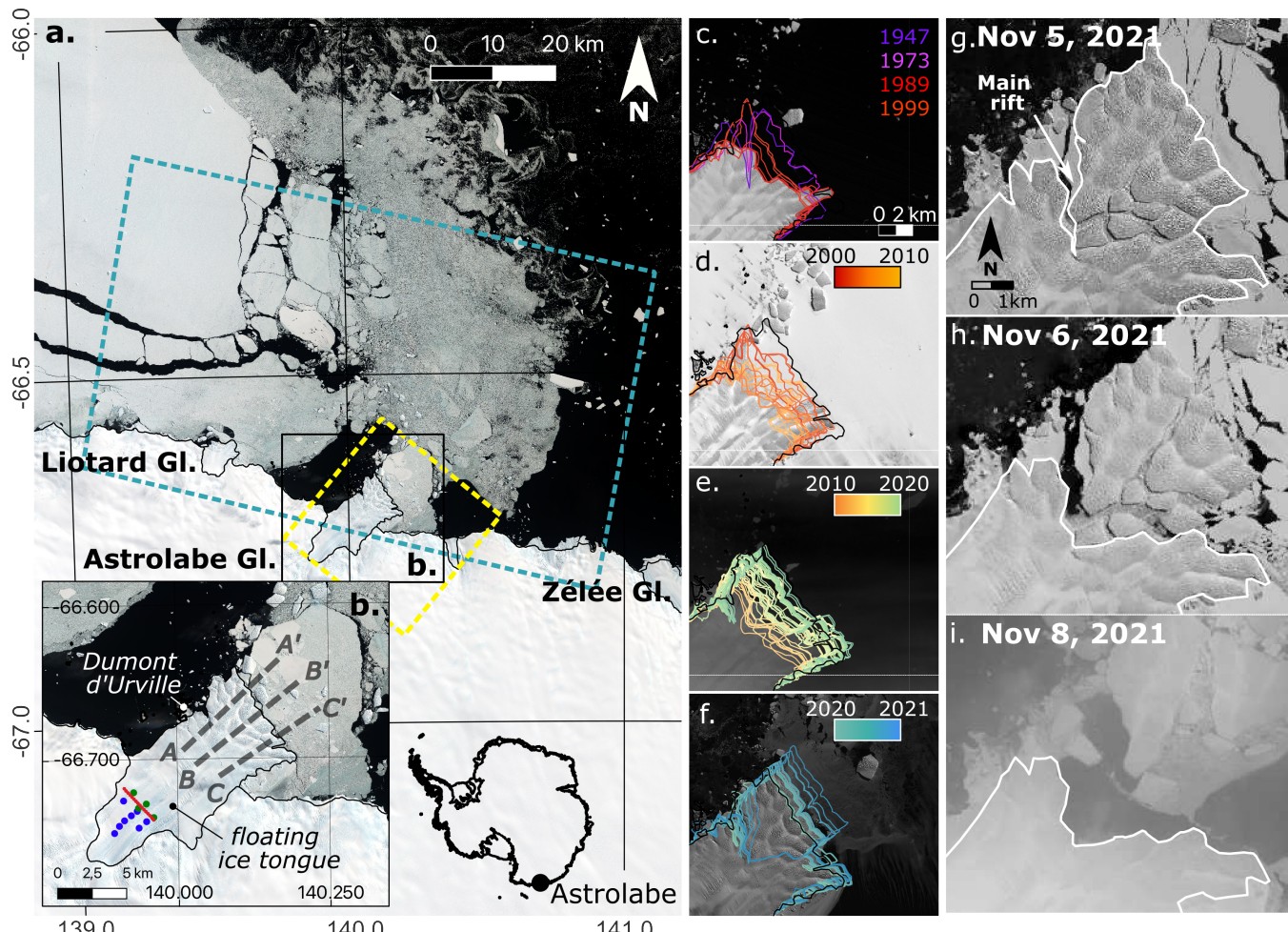

**Figure 1.** a) Location of Astrolabe Glacier with the coastline and grounding lines from Gerrish et al. (2022) and Sentinel-2 image of the February 7, 2020 in the background. The limits of the 4,000 km$^2$ box where the sea-ice extent is extracted is represented in dotted blue lines. Yellow dotted lines delineate the pixel extent and location of the sea-ice concentration grid from which the sea-ice concentration is extracted. The inset b) is a zoom over Astrolabe Glacier ice tongue and indicate the profiles where the evolution of the ice front position is presented (Figure 2). In-situ measurements are also represented: red dots for the location of the bamboo stakes, blue and green dots for the GNSS initial position of 2018 and 2021 campaigns respectively. Figures c) to f) show the ice front position at different dates. Figure g) to i) shows the calving of November, 6 2021 from the Sentinel-2 acquisition of November 5, 2021 (g), the Landsat-8 acquisition of November 6, 2021 (h), and the Sentinel-2 acquisition of November 8, 2021 (i).

the satellite imagery stack using the CO-REGIS algorithm (Stumpf et al., 2018), computes the displacement between pairs of co-registered images with the open source stereo-photogrammetric library MicMac (Rosu et al., 2015; Rupnik et al., 2017) and inverts the displacement time series using the TIO algorithm (Doin et al., 2011; Bontemps et al., 2018).

The Copernicus Sentinel-2 mission provides acquisitions of Astrolabe Glacier every three to six days, during the austral summer (September to April). In total, 59 Sentinel-2 images have been acquired over Astrolabe Glacier from February 2017 to early November 2021 without cloud cover. The pairing network is set so that each image is successively paired with the next five images, resulting in 280 pairs. Correlation is computed on a 5 by 5 pixel window using sub-pixel refinement. The displacement time series is inverted for each acquisition date with a spatial resolution of 1 by 1 pixel (i.e., 10 m x 10 m). The resulting displacement time series is interpolated at 30 days in order to compute the evolution of the ice velocity and to reduce the noise.

### 2.1.3 Computation of the strain rates from the ice velocity fields

Strain is a measure of how much a medium (here ice) stretches, compresses and deforms in all directions as it flows, whereas strain rates represent how quickly these deformations occur. Strain rates can therefore be computed using satellite-derived velocities (Alley et al., 2018; Cheng et al., 2021). We used the method described in Alley et al. (2018) and Nye (1959) to compute the longitudinal, transverse and shear strain rates using the annual estimate of the ice velocity derived from the GDM-OPT-ICE outputs (see section 3.2.1). Strain rates are computed at a spatial resolution of 20 metres.

## 2.2 In situ sensors

### 2.2.1 On-site GNSS observations and displacement measurements

A permanent GNSS network (https://astrolabe.osug.fr/) is maintained by the Institut des Géosciences de l'Environnement (IGE) on Astrolabe Glacier. It consists of 8 GNSS stations in 2018 and 4 stations in 2021 (mainly because of a lack of maintenance in 2019/2020 due to the cancellation of the summer operations in Antarctica because of the COVID pandemic; Figure 1b). The GNSS receivers and antennas are mounted on beacons specially designed to withstand harsh environmental conditions (e.g. strong winds, local wind-driven snow accumulation, ice movement, summer melting leading to beacon tilting or even collapse). These harsh conditions explain some gaps in the GNSS time series, especially during the Austral winters. The receivers are geodetic dual-frequency receivers (Trimble™NetR9) connected to Zephir geodetic antennas. The GNSS observations consist of 3 two-hour measurement sessions per day, where positions are averaged from 30-s sampling measurements. The positions are calculated for 24h measurements in PPP mode (Precise Point Positioning) using the GipsyX geodetic software (Zumberge et al., 1997). The accuracy is 1.5 cm (standard deviation 0.9 cm) in the horizontal component and 3.8 cm (standard deviation 2.7 cm) in the vertical component.

A field campaign carried out in 2020 to quantify the ice velocity in the vicinity of the grounding line position. It consisted of 16 bamboo stakes that were implanted in the ice during the winter of 2020 for one week between January 31, 2020 and February 7, 2020 (Figure 1b). The position of the stakes was measured on the first day and then, one week later, with a GNSS dual-frequency receiver, allowing the ice velocity to be estimated. The derived velocity is compared with the GNSS derived velocity from the 2018 and 2021 campaigns and to the NASA MEaSUREs ITS_LIVE (doi:10.5067/6II6VW8LLWJ7) available

in this part of the ice tongue. The result shows a good agreement between all datasets (Figure 2), meaning that the velocity remains locally constant in this part of the ice tongue over the years 2000 to 2018.

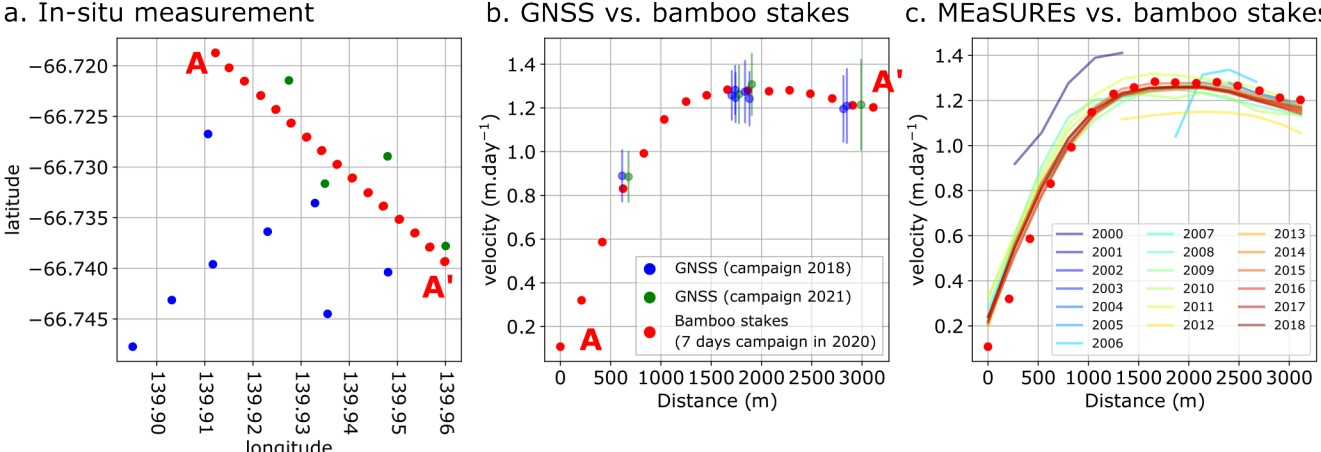

**Figure 2.** Comparison between yearly velocity measured with in-situ measurements from year-long GNSS campaigns and week-long bamboo stakes campaign: a) location of the GNSS's and bamboo stakes (see Figure 1 for the location on the ice tongue); b) comparison between the different in-situ dataset. c) comparison between the estimation of the velocity from the bamboo stake campaign and the yearly estimation of the velocity from satellite imagery from the NASA MEaSUREs ITS_LIVE (doi:10.5067/6II6VW8LLWJ7).

## 3  Results

### 3.1  Ice front position: 2000-2021

Changes in the frontal position are shown in Figure 3. The evolution of the ice front position varies from profile to profile. Between 1945 and 1995, historical images are sparse but show a maximum position of 4.2, 4.0 and 4.1 km for profiles AA', BB' and CC' respectively. In 2016 and 2019, the terminus of the ice front reaches this maximal position simultaneously for all three profiles. From 2019 to 2021, an unprecedentedly observed position of 7.2 km and 6.7 km is reached for profiles AA' and BB' respectively (Figure 3a, b). Conversely, the ice front position on profile CC' decreases progressively after 2020 due to successive calving events (Figure 3c). It should be noted that the central profile BB' regularly reaches its maximum position before the calving events of 2002 and 2010 (Figure 3b), while on profiles AA' and CC' the maximum position is reached only in 2002 or late 2002 (Figure 3a). From 2002 to 2010, the ice front position experiences annual calving periods of different length depending on the profiles considered: 2002-2008 for profile AA', 2004-2007 for profile BB' and 2003-2010 for profile CC'. A linear regression is performed to obtain the velocity of the ice front progression in between the successive calving events (Figure 3). The velocity varies greatly from one period to another, but it can be observed that the velocities are

significantly lower for profile CC' (1.17 m.day$^{-1}$-1.55 m.day$^{-1}$) than for profiles AA' (0.96 m.day$^{-1}$-1.79 m.day$^{-1}$) and BB' (1.75 m.day$^{-1}$-2.12 m.day$^{-1}$).

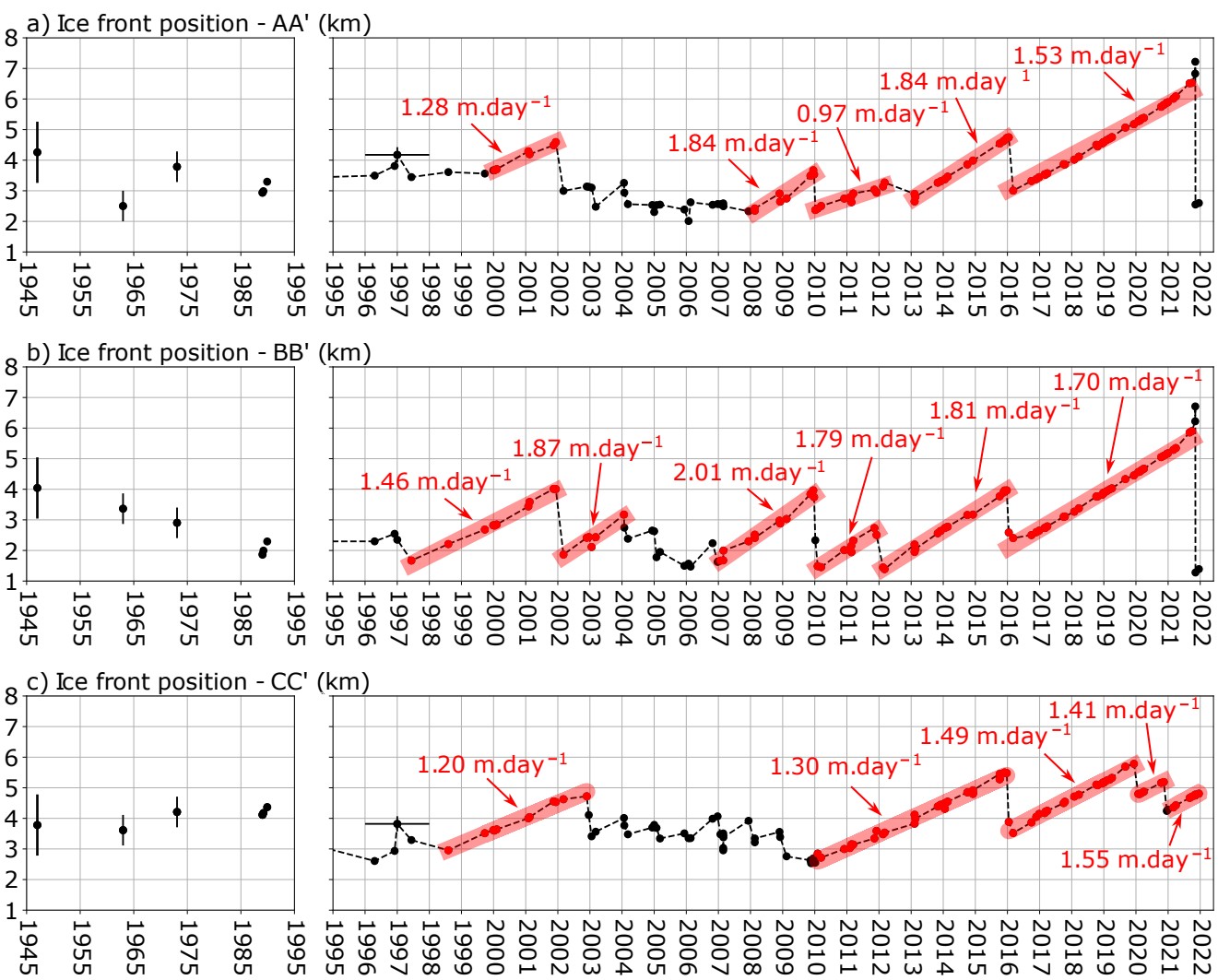

**Figure 3.** Evolution of the position of the glacier terminus along profiles AA' (a), BB' (b) and CC'(c); (see Figure 1b for the location of the profiles). The velocity of the ice front motion is indicated for the periods of ice front progression. The delineation of the terminus positions are mapped on Figures 1c-f.

## 3.2 Ice velocity: 2017-2021

The ice velocity is plotted for each year from 2017 to 2021 (Figure 4a) together with the derived longitudinal, transverse and shear strain rates (Figure 4c, d, e respectively). The yearly estimation obtained with GDM-OPT-ICE is compared to that

measured with in situ instruments (GNSS's and bamboo stakes campaigns). The in-situ data show that the velocity in this part of the glacier is very constant though time (Figure 2b, c) allowing for a comparison between different years. Figure 4b presents the ice velocity computed with GDM-OPT-ICE and the velocity measured with in situ instrumentation (i.e., GNSS's and bamboo stakes campaign). The comparison shows that the estimation of the velocity from GDM-OPT-ICE improves with time, with a poor accuracy in 2017 (RMS = 0.76 m.day$^{-1}$) and a much better one from 2019 (RMS < 0.25 m.day$^{-1}$). One can observe that the gradient of velocity from the western margin to the centre of the glacier is well retrieved with the GDM-OPT-ICE products of 2019-2021 (Figure 3b) while in 2017 and 2018, the limit between stable ice and the flowing ice tongue is retrieved in the wrong position with the GDM-OPT-ICE products. Indeed, in 2017, the GDM-OPT-ICE velocity of the 9 bamboo sticks located on the western side of the profile is almost zero for all locations. Conversely, a progressive increase of the velocity is measured during the bamboo sick campaign (Figure 2b) and by the ITS_LIVE products (Figure 2c). The same is observed in 2018, although the velocity derived from GDM-OPT-ICE is slightly larger than in 2017 (Figure 3b). The small number of cloud-free Sentinel-2 acquisitions for those years may explain the small RMS error for these two years, as well as the misestimation of the ice tongue boundary.

The velocity field shows a smooth gradient with lower velocity of about 1 m.day$^{-1}$ in the southeastern part of the glacier and a faster velocity of 1.2-1.5 m.day$^{-1}$ in the northwestern part of the glacier. In 2019, a small block of ice accelerated in the eastern part of the ice front (Figure 4a), which is also visible in the longitudinal strain rate field (Figure 4c, box B). This block disappears from both the 2020 mean velocity (Figure 4a) and strain rate field (Figure 4c) due to the calving of this part of the glacier in December 2020 (Figure 3d, profile CC'). In 2019, an extensive fracture appeared in the western part of the ice tongue in front of the Dumont D'Urville station, clearly visible in the longitudinal and shear components of the strain rate fields (Figure 4c, e; box A). The northwestern part of the ice tongue begins to exhibit higher velocities simultaneously in 2020 and 2021 (Figure 4a). In 2021, a complex network of localized increases of strain rates appears on the west northern part of the glacier delimiting the potential area of the future iceberg calving (Figure 4c, d, e). This complex network delineates the fractures that were observed on the ice in the first available summer acquisition in September 2021 and which remained unchanged until the ice calving (Figure 1f). In addition to the evolution of the fractures in the ice, one can also observe the high strain rates (> 0.002 day$^{-1}$) that are clearly identifiable along the lateral boundaries of the glacier over time (Figure 4e).

### 3.3   Ice tongue break off: 2021

The displacement time series is linearly interpolated with a time step of 30 days from the first acquisition in 2017 to November 5, 2021. Mean monthly velocity and strain rate fields are derived from this interpolation. We examine the evolution of the strain rates for the period January and November 2021 to understand the dynamics of the recent calving (Figure 5). The strain rate maps show a high concentration of strain localized along linear structures that grow progressively in length from April 2021 to November 2021. We set a threshold on the strain rates in order to analyse the evolution of these localized strain concentrations, as well as the occurrence of the spatial connection between them (Figure 5a). The evolution of their growth is complex, with transitions from one component to another. For example, the main rift exhibits a clear longitudinal strain rate from April 2021 to September 2021 which evolves into a shear strain rate in October-November 2021 (Figure 5a). From May 2021, a

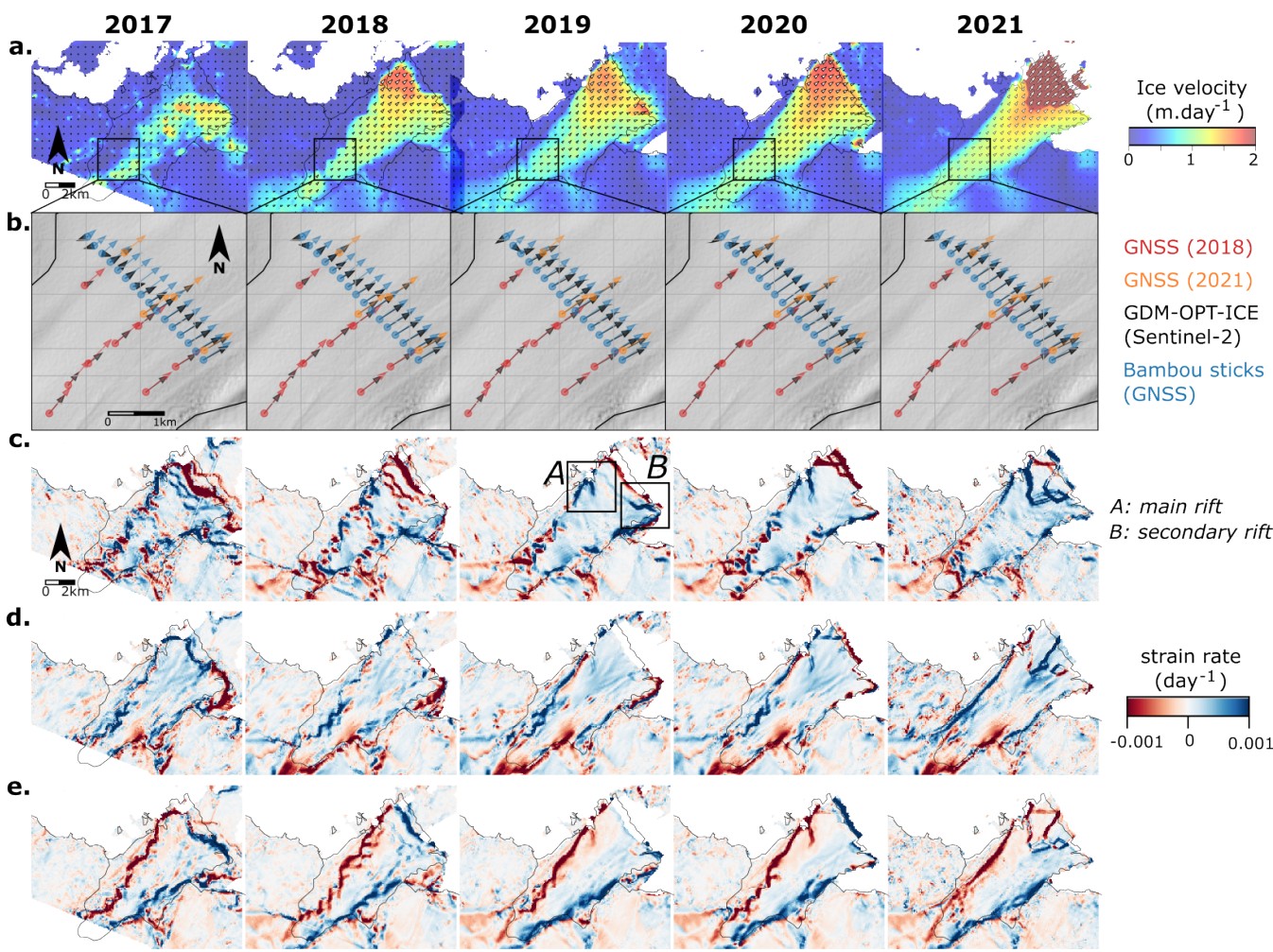

**Figure 4.** a) Yearly estimation of ice velocity for Astrolabe Glacier for the period 2017-2021, b) comparison of the velocity magnitude and direction as measured by the in situ instrumentation (GNSS's and bamboo sticks campaign) and as measured by GDM-OPT-ICE. Figures c, d, e present the longitudinal, transversal and shear strain rates derived from the ice velocity fields.

large concentration of strain appears in the transverse component along a north-east/south-west trending fracture (Figure 5a). Similarly, a third fracture appears in the longitudinal component on the eastern side (Figure 5a). These fractures grow rapidly and join in June 2021 (Figure 5a). One can also observe that from October 2021, most of the fractures show a shear strain rate, probably due to the rotation of the blocks. We then analyse the Sentinel-1 SAR images from May 2021 to August 2021 in order to validate these observations. We observe that the fracture network suddenly opened between June 13, 2021, and June 25, 2021 (Figure 5b, c), which is consistent with the timing of the connection derived from the strain rate time series (Figure 5a). It can be noted that compressional strain rates are measured from 2017 to 2020 at the terminus of the glacier tongue with an absolute value greater than 0.001 day$^{-1}$ while they are not observed in 2021 (Figure 4c).

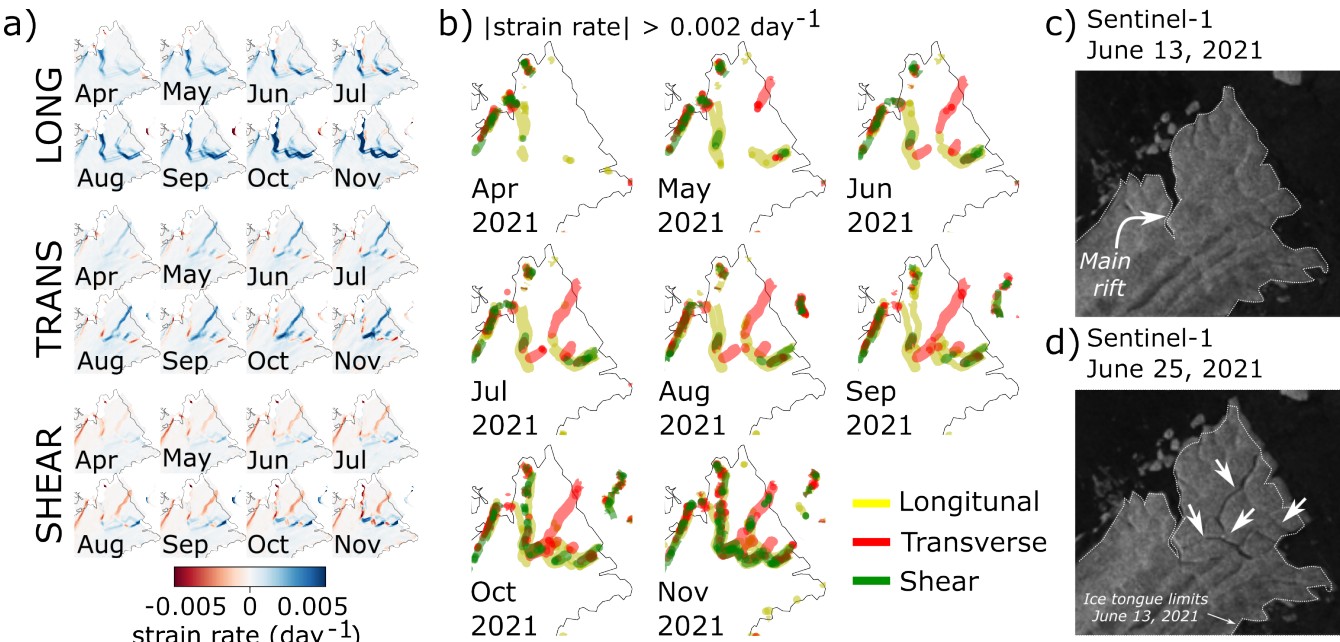

**Figure 5.** a) Evolution of the strain rate from April to November 2021 in the longitudinal, transverse and shear direction. b) Mapping of the area where the strain rates are larger than 0.002 $\text{day}^{-1}$ from April to November 2021. The three strain rate components (longitudinal, transverse and shear) are plotted together with different colours. Subsets c) and d) are showing the occurrence of the fractures detected with Sentinel-1 acquisitions of June 13 and June 25, 2021. The white arrows indicate the location of the main rift (c) and of the network of secondary fractures appearing in June 2021 (d).

### 3.4 Sea-ice forcing

We analyse time series of sea-ice extent and concentration in the region of Astrolabe Glacier (Figure 1a). The data for sea-ice extent and concentration are downloaded on the NSDIC repository (Fetterer et al., 2017) and cover the periods from 1979 to the end of 2021. We cropped the data to analyse the monthly variation of the sea-ice extend over an area of 4000 $\text{km}^2$ around Astrolabe Glacier (dashed blue lines on Figure 1a). This area is arbitrarily chosen to represent the influence of regional variation in sea ice conditions. The daily variation in the sea-ice concentration is taken at the pixel (25 km x 25 km)

encompassing Astrolabe Glacier (dotted yellow lines on Figure 1a) to focus on the conditions at the Astrolabe ice tongue.

We observe a significant change in the periodicity of sea ice around Astrolabe Glacier in the last decade (2011-2021). From 1979 to 2011, the extent of the sea ice decreases significantly every year during the summer (Figure 6a). From 2011 to 2021, the annual disappearing of sea ice does not occur every year (Figure 6a). Indeed, during two consecutive periods: 2012-2016 and 2016-2021, the extent of the sea ice remains maximum during the summer (Figure 6a). In detail, one can see

that during those two periods, the sea-ice extent can decrease occasionally (e.g., early 2015, 2018) or during larger periods such as for the 2018-2019 austral summer (Figure 6a). However, the reduced length or absence of sea-ice-free periods during

2012-2016 and 2016-2021 is notably different from the previous decades. The time series of daily sea-ice concentration shows similar observations (Figure 6b). Before 2011, the sea-ice concentration drops below 15% for periods of 2 to 3-4 months from November to mid-March, with small variations in the length of sea-ice-free periods (Figure 6b). From 2008 to 2011, one can observe a decrease in the duration of low sea-ice concentration to 2 months (Figure 6b, d), corresponding to a shift in the onset of the sea-ice-free period from November to mid-December/January, while the end of the sea-ice free period remains stable over time: early to mid-March. From 2012 to 2016, the sea-ice-free periods disappear or are shortened to less than one month (i.e. February 2015; Figure 6d). From 2016 and 2021, the regime of sea-ice concentration is highly variable, with years with no to very short periods of sea-ice-free conditions (austral summer 2016-2017, February 2020; Figure 6b, d) and years with prolonged sea-ice-free conditions (December 2018-March 2019). From austral summer 2020-2021, the duration of sea-ice-free conditions seems to return to the pre-2012 level, with a duration of 3-4 months from mid-November to March (Figure 6b, d).

We compare the evolution of the sea-ice extent and concentration with the evolution of the ice-tongue area (Figure 6c). We observe that the periods 2012-2016 and 2016-2021 corresponds to periods of significant extension of the Astrolabe ice tongue (Figure 6c) with an increase of 15 km$^2$ between 2012 and 2016 and of almost 20 km$^2$ between 2016 and 2021. For the period 2002-2012, the ice tongue extension is much more limited due to the regular calving at different locations on the ice front (Figure 3). Before 2002, satellite imagery is scarcer, but the ice tongue appears to have reached a fairly advanced position in 2002, with an area of almost 81 km$^2$. This advance cannot be linked to significant variations in the sea ice seasonal cycle. We report the calving events observed in the analysed satellite images (Figure 6d) with the uncertainty in the date of the different calving events. One can observe that all detected calving occur when sea-ice concentration decreases at the end of the Austral autumn (Figure 5d), except for Austral summer 2006-2007 where several calving events are reported and do not occur necessarily at the onset of the decrease of sea-ice concentration.

## 4  Discussion

To understand the recent evolution of Astrolabe Glacier, we investigated the evolution of sea ice in the vicinity of the Astrolabe ice tongue. The evolution of sea ice, and in particular, landfast sea ice, is usually assumed to delay the break-up of ice tongues and favour its extension by buttressing the ice tongues and protecting them from ocean swells (Massom et al., 2010, 2018; Rott et al., 2018; Gomez-Fell et al., 2022). At Astrolabe Glacier, we observe a significant change in the periodicity of sea ice in the last decade (2011-2021) compared to the previous observations (1979-2011; Figure 6). The recent multi-year periods of sea ice are well correlated with the spatial extension of the Astrolabe ice tongue (Figure 6) and seem to validate the assumption that sea ice protects the ice tongue and favours its extension. Furthermore, the disappearing of such protection has been reported to trigger crack propagation, in some cases leading to rapid calving (Miles et al., 2017; Cassotto et al., 2021; Gomez-Fell et al., 2022; Christie et al., 2022). At Astrolabe Glacier, we also observe that the calving, when it occurs, systematically takes place at the beginning of the sea ice disappearing (Figure 6d) which seems to confirm the potential of sea ice disappearing as a triggering factor for calving. However, the analysis of satellite images from 2017 to 2021 at the Astrolabe ice tongue shows that the rifts or crevasses that lead to the 2021 calving event form several months to years before calving (Figure 4, 5), suggesting

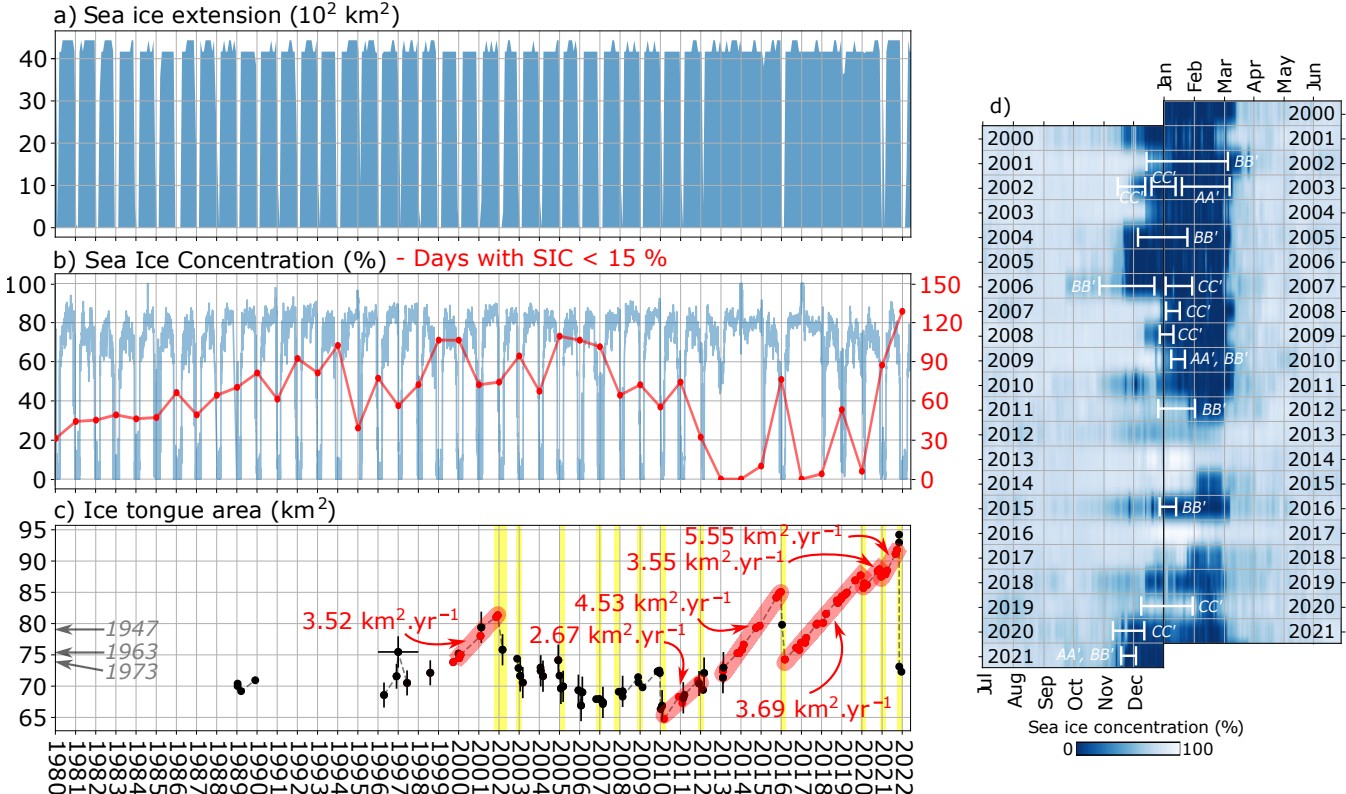

**Figure 6.** Evolution of monthly sea-ice extent (a), daily sea-ice concentration and number of days with sea-ice concentration lower than 15% (b). Monthly sea-ice extent is computed for a wide region of 4000 km$^2$ around Astrolabe Glacier (dotted blue line in Figure 1a) while the daily sea-ice concentration is taken for the pixel of 25 km by 25 km at the Astrolabe ice tongue (dotted yellow line in Figure 1a) and both are extracted from Fetterer et al. (2017). The evolution of the ice-tongue area is presented in (c) with in red the period of extension and the ice tongue growth speed. Calving event that could be observed with satellite imagery are plotted in yellow. In years 2003-2005, calving likely occurred, although no observation can confirm the date. Figure d) also presents the evolution of the daily sea-ice concentration from 2000 to 2021 and calving events are reported.

a different mechanism. The presence of rift and fracture networks in the ice tongue several years to several months prior to calving has been reported in other glaciers (Fricker et al., 2005; Walker et al., 2013, 2015; Cheng et al., 2021; Larour et al., 2021; Gomez-Fell et al., 2022). In most cases, rift growth or (re-)activation is observed during the Austral summer (Fricker et al., 2005; Walker et al., 2013; Cheng et al., 2021; Gomez-Fell et al., 2022) a and little rift propagation is reported during the Austral winter (Walker et al., 2013; Larour et al., 2021). On the Astrolabe Glacier ice tongue, the main rift is located in front

of the Dumont D'Urville research station and was initiated in 2019, a year with almost two consecutive months of low sea-ice concentration. The absence of a significant rift in 2017 and 2018 suggests that sea ice have had the effect of inhibiting rift growth on the Astrolabe ice tongue and delaying calving in 2017-2018 and, possibly, in 2012-2016. We note that no calving occurred in 2019 despite the long period of low sea-ice concentration while in 2020, the eastern part of the ice tongue calved

at the onset of the short (one month) period of sea-ice concentration decrease (Figure 6d). These observations suggest that sea ice acts as a glue to hold the ice tongue together, and that deep opened rifts/cracks must be present for calving to occur.

In June 2021, we observe the sudden opening of a complex network of fractures in the middle of the Austral winter (Figure 5). Larour et al. (2021) proposed a mechanism to explain the winter propagation at the Larsen C Ice Shelf, Antarctica, prior to the calving of iceberg A68, based on the critical thinning of the ice shelf and of the ice mélange within the rifts. Here, the critical thinning of the ice tongue due to its exceptional extension may explain this timing (Robel, 2017; Larour et al., 2021; Åström and Benn, 2019) although it would likely favour the propagation of the rift along the same direction as the pre-existing rift, which is not observed at Astrolabe Glacier (Figure 5). Instead, the main fracture propagating in June 2021 is oriented along the flow direction and opened in extension (Figure 5a, b). Another possibility to explain the development of these fractures could be a transition from a ductile to a brittle behaviour with the decrease of temperature during winter that may favour fractures along the flow resulting from the differential compressive load due to sea ice buttressing and rift opening (Figure 4). This mechanism might be possible as the compressive longitudinal strain seems to disappear in 2021 at the glacier terminus (Figure 4c) and the rift opens progressively. However, such a scenario remains to be validated because it would keep the terminus of the ice tongue in the same position due to the effect of sea ice buttressing, which is not observed (Figure 4c, d), and because the compressive strength of the ice is much higher than the extensive strength (Benn et al., 2007). The presence of extensive circumferential stress which occurs when the unconfined part of the ice tongue reaches a certain extension (Wearing et al., 2020) should be also considered, as well as the presence of basal channels and basal melt that may play a role in the dislocation of the ice tongue (Vaughan et al., 2012; Alley et al., 2023). The difference in calving cycle and ice velocity between the eastern and western parts of the glacier terminus also suggests that bathymetry beneath the ice tongue controls the location and evolution of the rifts. Our analysis remains limited, and further modelling is needed to understand the mechanisms that lead to the apparition of these fractures at this time of the year (Åström and Benn, 2019; Crawford et al., 2021; Alley et al., 2023).

Sea ice is strongly linked to regional and local atmospheric and oceanic conditions (Fogt et al., 2022). At the continental scale, records in Antarctica show a positive increase of the sea-ice extent from 1979 to 2016 with a minimum of global sea-ice extent recorded in summer 2017 (Fogt et al., 2022). In the Astrolabe region, which includes the Adélie Coast and George V Land, the calving of the Mertz Ice Tongue in 2010 (Massom et al., 2018) led to strong changes in sea ice production and location, reflecting regional changes in the oceanic and atmospheric currents (Campagne et al., 2015). The Mertz Ice Tongue was hit by the B09B iceberg in 2010 and lost about 80 km of its length (Massom et al., 2015). Campagne et al. (2015) shows that calving of the Mertz Ice Tongue is directly responsible for a 50% increase in sea ice concentration in the Mertz Glacier polynya. The Mertz Ice Tongue probably acts as a barrier to westward ice advection within the Adélie Depression (Lacarra et al., 2014; Campagne et al., 2015) and its regular calving is followed by decades of high sea-ice concentration before the tongue regains sufficient length (Campagne et al., 2015). The region of Astrolabe Glacier, some 230 km to the west, appears to be undergoing similar changes (Figure 6), and the calving of the Mertz Ice Tongue is likely at the origin of the transition in the sea ice seasonal cycle at Astrolabe Glacier (Figure 6). Moreover, Miles et al. (2022) reports similar observations further west on the Adélie Coast, with the continuous growth of the Commandant Glacier (Adélie Coast) from 2010 to 2018 due

to the presence of persistent sea ice. This illustrates how a calving event such as the Mertz Ice Tongue calving in 2010 can significantly alter the calving cycle of neighbouring ice tongues several hundred kilometres away, which remains difficult to account for in current models (Edwards et al., 2021; Miles et al., 2022). The extent of the regional impact of the 2010 Mertz Ice Tongue calving is not clearly known, as most studies focus on the Georges V land area and the Adélie Depression (Kusahara et al., 2011, 2017; Campagne et al., 2015; Cougnon et al., 2017). In addition, the evolution of the Adélie Coast and Georges V land ice shelves remains limited (Frezzotti et al., 1998; Frezzotti and Polizzi, 2002), preventing a better understanding of the environmental forcing (Massom et al., 2018; Christie et al., 2022).

## 5  Conclusions

In this study, we analysed the evolution of Astrolabe Glacier located in Terre Adélie/Adélie Coast, Antarctica. We used open access optical satellite imagery (MODIS, ASTER, Landsat and Sentinel-2) completed by ERS and RADARSAT images to map the evolution of the ice front from 1947 to 2022. We also measure the surface velocity and derived strain rate fields between 2017 and 2022, using image correlation of Sentinel-2 images. The recent evolution of the glacier shows an unprecedentedly documented extension of 95 km$^2$ favoured by the concomitant high concentration of the sea ice in the region during 2011-2021 in comparison with previous records (2000-2011). Early sea ice melt in November 2021 favoured the release of a 20 km$^2$ iceberg in the northwestern part of Astrolabe Glacier. This is the first time that a calving of this size has been documented on Astrolabe Glacier. We also observed that a complex network of fractures opened during the austral winter of June 2021, several months before the iceberg calved. This study demonstrates the importance of time series of ice velocity and strain rate fields derived from high resolution optical satellite imagery in documenting fracture opening and raises further questions about the mechanism of fracture propagation.

*Data availability.* We acknowledge the use of imagery from Copernicus Sentinel-1 and 2 data (https://dataspace.copernicus.eu/), NASA MODIS, ASTER and Landsat images (through https://earthexplorer.usgs.gov/). The GNSS observations are accessible on the Astrolabe repository: https://astrolabe.osug.fr/. The Astrolabe sketch from the US Navy Operation Highjump aerial photographs of the Adélie coast is available on https://archives-polaires.fr/idurl/1/14865). We also acknowledge the use of NASA MEaSUREs ITS_LIVE doi:10.5067/6II6VW8LLWJ7, and of the NSIDC Sea Ice Index data set : https://nsidc.org/data/seaice_index. The data processed in this study is available on https://geohazards-tep.eu/ and upon request.

*Author contributions.* FP designed the experiments with contributions from DZ, ELM, JPM and CH. ELM provided the GNSS data and JPM processed them. FP processed the satellite data. All co-authors participated in the writing and/or revision and approval of the submitted manuscript

*Competing interests.* We have no competing interests.

*Acknowledgements.* The GDM-OPT-ICE service is developed and maintained by ForM@Ter (Data and Service for the Solid Earth: en. poleterresolide.fr) and exploited on the EOST/A2S High Performance Computing (HPC) infrastructure of University of Strasbourg (1.5 Tier Mesocentre) allowing optimized computation. The service is accessible on-demand through the ForM@Ter data hub (en.poleterresolide.fr/services-en/gdm-en/#/optic) and the Geohazards Exploitation Platform (GEP: geohazards-tep.eu). The authors also acknowledge the support of the French Agence Nationale de la Recherche (ANR), under grant ANR-20-CE01-0006 (project HighLand). The GNSS data have been collected with support from the French Polar Institute (IPEV) as part of project #1053 DACOTA. The authors thank the two anonymous reviewers for their constructive comments that helped improve the manuscript.

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
