# Peer review of "Surface dynamics and history of the calving cycle of Astrolabe Glacier (Adélie Coast, Antarctica) derived from satellite imagery"

_EGUsphere, 2023_

## Author Comment (AC1)

Dear Editor, dear reviewers,

We thank warmly the reviewers for the careful reviews and for their comments. We propose a new version of the article taking into account the remarks of the reviewers. We explain in details the reason of our choices.

Sincerely,

Floriane Provost, on behalf of all co-authors,

NOTE: In the following document, the referee comments are in normal fonts and the answers are in blue font.

**

**Reviewer 1 comments:**

This manuscript reports on the recent calving dynamics of Astrolabe Glacier, a small glacier in Terre Adélie Land, East Antarctic. The manuscript reports of an anomalous advance of the ice tongue which the authors link to strong landfast sea-ice conditions, but they also highlight that fractures can develop during the winter. Sea-ice buttressing is a potentially important but understudied topic, and the detailed inter-annual observations presented in this manuscript have the potential to be useful for the wider community. However, I think major revisions are required here before publication can be considered. My concerns are detailed in the below comments:

**Major comments:**

**"Unprecedented advance"**: The manuscript states in a few locations that there was an unprecedented advance of the ice tongue up to November 2021. This may well still prove to be correct, but I cannot be certain of this because the study does not show the full time series of available satellite imagery. The manuscript looks in detail between 2000-2022, but what about the earlier satellite imagery? I am pretty sure there is available imagery going back to the 1970s e.g. 1997 RADARSAT mosaic, 1992 ERS data, Landsat4/5 late 1980s, Landsat 1 early 1970s, maybe even the ARGON mosaic from 1963 (Kim et al., 2007) or even aerial photography from the 1940s. I have not checked for each of those satellites, but there is certainly a cloud-free Landsat-1 image from the early 1970s. I appreciate that there are gaps between these images, but they are still important to include to help determine if this really is an unprecedented advance. The ice-front extent from these images should be added to Figure 5a.

We would like to stress that we stated "unprecedentedly **observed** extension" which implies "in the limit of the data considered". We never state that the observed extension in 2021 is for sure unprecedented, and we wrote it in the discussion: "Due to the limited temporal extent of the datasets and sparse

availability of the satellite imagery before the 2000s, it is impossible to conclude if these changes are exceptional or part of a longer cycle" (L230).

The Astrolabe ice front evolution has not been documented since Frezzotti et al., 2002 (this analysis spread from 1947 to 1989). Our goal was to resume this work looking at satellite optical acquisitions freely accessible at high spatial resolution ($< 15$ m) to distinguish the ice front from the sea-ice or landfast sea-ice. Concerning the datasets mentioned by the reviewer:

- **Radarsat**: we do not have access to raw RADARSAT images. We map the ice front from the RAMP product [Jezek and Barry., 2013] and added it to the dataset.

- **ERS data**: ERS data are available over the Astrolabe from 1996. We mapped the front position from 1996 to 1999 in order to fill the gaps in this period.

- **Landsat 1**: there is only one cloudless image available from Landsat-1 on January 29, 1973.

- **Landsat 4/5**: there are 3 cloudless Landsat 4/5 acquisitions in 1989.

- **ARGON mosaic 1963**: The ice front position is very difficult to determine from this dataset at the Astrolabe glacier due to bad contrast of the acquisition. We attempt to map the ice front with this acquisition (Figure 1b, revised).

- **Aerial photography**: in 1947, one campaign was performed and sketched maps were derived (https://www.archipoles.com/idurl/1/14865). We digitize the sketched map and perform a first order coregistration in order to map the ice tongue ice front (Figure 5a). It should be noted that we observe high distortion in the sketched map, but the trend of our mapping is very similar to [Frezzotti and Polizzi, 2002]. We do not have access to more recent aerial photography.

We also added ice front position from MODIS images from 2000 to 2010 in order to complete the gaps in this period with error bars corresponding to an error of $\pm$ 1 pixel in the ice front position. We updated all figures accordingly. In conclusion, it confirms what we stated: for the considered dataset (even extended in the revised manuscript) the extension of the Astrolabe glacier's shelf in 2021 has never been observed before.

**Sea-ice concentrations:** As I understand, the authors extract mean sea ice concentrations from a 4,000 km box surrounding the glacier, they infer periods of multi-year landfast sea-ice conditions from this analysis. I am not sure this is the case. The dataset the authors use is not the best for determining Landfast sea-ice conditions because the pixels are course and they do not extend all the way to the coastline.

In the new version of the paper, we plot the spatial extent of the box (80 km along the coast x 30 km across the coast) surrounding the glacier that was used for extracting the sea-ice extent. The box comprises the ice tongue (Figure 1 revised) and we checked before the analysis that the sea-ice product encompassed the glacier (it does) and that the coastline was not defined on the glacier (it is not). We chose this dataset because it spans a long time frame: 1979-2021 including the period of the 2021 calving. We assumed that if we look at the coastal areas or to the direct surroundings of the glacier, the sea-ice is likely to be fastened to the coast.

[Figure]

**Revised Figure 1**.

In order to validate this assumption, we analyzed another product from [Fraser et al., 2020] specifically constructed to map landfast sea-ice but with a more restricted time period: 2000-2018. Figure 1 (below) shows the comparison between the two products for the same area: a) the 4,000 km$^2$ box and b) for the pixel area surrounding the glacier (Figure 2).

First, one can observe that the [Fetterer and Windnagel, 2017] products do not exhibit major differences between the two considered areas. Secondly, at the vicinity of the glacier, the detection of landfast sea-ice from [Fraser et al., 2020] is in agreement with the general trend of [Fetterer and Windnagel, 2017] (Fig 1b) in terms of absence/presence of landfast sea-ice or sea-ice. The only major

[Figure]

Figure 1: Comparison between the extension of sea-ice from [Fetterer and Windnagel, 2017] and the extension of landfast sea-ice from [Fraser et al., 2020] for the two areas: a) the 4,000 km$^2$ box (in blue on Figure 2), and b) for the pixel area surrounding the glacier (in yellow on Figure 2).

difference is for year 2013 (Fig 1b) where the [Fetterer and Windnagel, 2017] detects the presence of sea-ice while [Fraser et al., 2020] shows landfast sea-ice free conditions. This may be explained by the difference of spatial and temporal resolution of the two datasets and, by the fact that during January-February 2013 an unusual polynya appeared at the Astrolabe glacier (Figure 2). It seems that the direct surroundings of the Astrolabe ice tongue remained partly trapped in landfast sea-ice during 2013 (Figure 2).

We do agree that a more detailed (at finer spatial resolution) analysis could be conducted in the future to map precisely the type of sea-ice (i.e. ice mélange, polynyas, landfast sea-ice, icebergs, etc.) surrounding the Astrolabe glacier. However, we do believe that the considered product ([Fetterer and Windnagel, 2017]) captures the general trend of landfast sea-ice conditions at the Astrolabe glacier.

Indeed, a quick look at the MODIS imagery shows that actually the glacier is sea-ice free for large portions of nearly every summer, including the period of ice tongue advance. So the claim that multi-year fast ice has forced the advance of the ice tongue is incorrect. It appears that there is no MYFI at any stage over the period 2011-2021. I think a useful figure for any revised manuscript could be a multi-year panel showing a MODIS image showing the sea-ice conditions at the sea-ice minima (roughly March) every year from 2000 (or at least the period of anomalous advance 2011-2022), this would enable the reader to better visualize the sea-ice conditions. (see MODIS imagery on NASA Worldview website; link below).

[Figure]

Figure 2: MODIS acquisition of January 10, 2013 centered on the Astrolabe glacier. The acquisition shows the polynya that developed in January-February 2013 at the Astrolabe glacier. The blue square represents the extent of 4,000 km$^2$ box used to extract sea-ice extent. The yellow square represents the extent of the pixel of the [Fetterer and Windnagel, 2017] product located on the Astrolabe glacier. The area of the pixel is around 600 km$^2$.

We thank the reviewer for the link to the MODIS viewer. As requested, please find the time series of early-summer (corresponding to late austral summer) MODIS images from 2001 to 2021 in Figure 2. We looked for and show the minimum of sea-ice extent we found for each year.

One can observe that the first two rows (2001-2010) show clear sea ice free conditions every year. Conversely, on the two next rows (2011-2020), the area surrounding the Astrolabe glacier is always occupied totally or partially by sea ice, except for years 2011, 2016 and 2021. In particular, in 2014, 2015, 2017, 2018 it is clear that sea-ice attached to the glacier is visible. This is as well confirmed by the dataset of [Fetterer and Windnagel, 2017] and [Fraser et al., 2020] and by the data presented in Figure 5a.

The definition of Multi Year Fast ice can be defined as "first-year", "second-year" and "multi-year" sea ice [Fraser et al., 2023]. However, [Fraser et al., 2023] is also stating: *"Due to the coarse resolutions of gridded products (e.g., Melsheimer et al., 2022) it is difficult to discriminate between second-year and multi-year*

[Figure]

Figure 3: MODIS images at the Astrolabe glacier ice tongue for period 2000 to 2021. One image per year is displayed, corresponding to the observed minimum extent of sea ice.

*ice. Hence only the latter term will be used herein."*

The screening of all the dataset shows that landfast sea-ice around the Astrolabe does not disappear for two consecutive years in 2014-2015 and in 2017-2018. Hence, we can conclude according to the definition that these periods are MYFI. MYFI is also visible on the [Fraser et al., 2020] product on the period 2014-2015 and 2017 (year 2018 is not available). Hence, we strongly disagree with the reviewer, and we would like to kindly ask the reviewer to indicate precisely the sea-ice periods he/she is referring to as sea-ice free for these years.

Moreover, we derived the main two periods of sea-ice extension from Figure 5a, which is based on monthly analysis of the sea-ice extent. We already stated clearly in the text that these two periods correspond to MYFI **or** very short periods of sea-ice free conditions: "in the period 2011-2021, multi-year landfast sea-ice occurred with no sea-ice free conditions, or very short and/or episodic periods of sea-ice free conditions (Figure 5b) at the vicinity of the Astrolabe ice

tongue." (L238-240). We do not mean here that the entire period 2011-2021 is a MYFI.

This is important, the other studies mentioned in the manuscript (Miles et al., 2017 – Porpoise Bay, Gomes-Fell et al., 2020 – Victoria Land and Christie et al., 2022 – Antarctic Peninsula and other studies) link the advance of outlet glaciers to persistent multi-year fast ice, and when the fast ice finally breaks away the ice shelves/tongue calve instantly. A different process is clearly operating here and the differences between the studies mentioned above and the processes operating here should be discussed in any revised version. The increased sea-ice concentrations in the 4,000 km box are clear in the summer post 2011 and Figure 5b is convincing, but the mechanism that the authors hypothesize is driving the changes to the ice tongue needs to be much more clear.

We improved the discussion and attempted to be clearer in the similarities and differences between the cited studies and glacier sites. We observe that the presence of persistent sea ice fasten or at the vicinity of the Astrolabe glacier ice tongue corresponds to period of glacier extension. It remains difficult to prove if that is due to the effect of the buttressing of the sea ice, to the protection it offered against ocean swell, or most probably both. However, this is in line with Miles et al., 2017, Gomes-Fell et al., 2020 or Christie et al., 2022. Conversely, we do not observe a simultaneous fracturing of the ice tongue during the onset of the sea ice free period. We do observe that at the Astrolabe, the presence of fractures prior to sea ice disappearing is key to initiate ice calving, and that sea ice unbuttressing does not generate nor accelerate fissure/rift opening. The Astrolabe calving of 2021 present similarities with the one of Larsen C-A68 of 2017 with fissures propagating during the austral winter (of 2021; [Larour et al., 2021]). [Larour et al., 2021] showed that the thinning of ice mélange within rifts can lead to the acceleration of the rift opening. At the Astrolabe, the sea ice surrounding the ice tongue and within the rift partially melted in 2020, and it is possible that the renewed sea ice was thinner and weaker than prior to 2020 which would have favored the opening of the rift [Larour et al., 2021]. However, we do not observe an opening of the main rift but rather a dislocation of the entire western part of the ice tongue front during austral winter 2021. We hypothesize that this could be due to both the action exerting by the opening of the rift and the resistance of the sea-ice buttressing resulting in a compressional stress within the western part of the ice tongue and hence the opening of fissures in the transverse direction. This hypothesis remains to be tested with a model.

It is also essential that the authors detail the sea-ice concentrations in the methods section of the paper, it has been completely missed out. Including where you have actually extracted the data i.e. show the 4,000 km box on a figure.

We added the bounding box and the pixel location and extent where the sea-ice area and concentration are extracted (Revised Figure 1). We are using an external dataset [Fetterer and Windnagel, 2017], and we do not think one specific paragraph is needed to describe this data. We added a paragraph in the discussion about their quality and possible processing to monitor more accurately the sea-ice around the Astrolabe.

**Glacier acceleration and ice break off: 2021:** In general I was a little confused in this section, are you refereeing to an acceleration of the entire glacier? Or is this just a local acceleration on the ice tongue near the ice-front?

We refer to the frontal part of the glacier where the network of fractures appears in austral winter 2021. We added a sentence in this section to be more specific.

**Title**: I feel it could be made more impactful if it included something related to sea-ice, the process that the authors hypothesize is driving change here. Very few glaciologists would recognize Astrolabe Glacier and I fear most would simply skip over the manuscript based on the current lengthy title. I think the title needs to include something relating to sea-ice to entice a much wider readership. Perhaps something along the lines of: 'Anomalous advance of the Astrolabe Glacier Tongue driven by more persistent sea-ice conditions 2011-2021'. (Or something similar)

We thank the reviewer for this suggestion.

**Cause of shift in sea-ice conditions:** The shift in sea-ice conditions appears is around 2011. This more or less coincides with the calving of the Mertz Glacier tongue, just round the corner from Astrolabe. The calving of the Mertz Glacier fundamentally changed the local sea-ice conditions (see Tamura et al., 2012; Campagne et al., 2015 and several others). Feel free to ignore, but maybe worth briefly investigating and would make an interesting discussion point.

We agree and added a paragraph on the discussion about the influence of the Mertz glacier tongue on the region. We briefly address this point, as we do think it would require a regional investigation and additional dataset to discuss in depth the influence of the Mertz calving on the Astrolabe glacier calving cycle.

**Figures**

Figure 1: There is no satellite image of the entire glacier at any point in the manuscript. This is an essential requirement and it would make sense for this to be in figure 1. I would recommend replacing the DEM with a Landsat or Sentinel-2 image. You could also add all the mapped ice-fronts to this figure.

We thank the reviewer for this suggestion and modified figure 1 accordingly.

Figure 5: I think panel (a) is a nice figure, but you could consider extending the ice-front position change back to the 1970s. It is difficult to judge panel

(c) because I cannot really distinguish any changes from the color scheme, please consider amending this.

We maintained the time limits of Figure 5a but updated the figure with the additional ice front position described in the answer of the first major comment.

New figure suggestion: A multi panel figure showing a MODIS image of the glacier every sea-ice minima (roughly e.g. Late Feb to March)

Here, we do not agree that this figure is pertinent to the article. Although it provides visual context, it is not representative of the complex sea-ice cycle.

**Minor comments**

Numerous incidents of Figures or references not bracketed

We carefully reviewed the manuscript to correct this.

Line 19-23: Not sure, I buy this argument. Most of the ice shelves in West Antarctica are actually quite small. There have also now been at least a dozen of East Antarctic focussed studies on individual glaciers. Please revise.

We change the sentence for: "The monitoring of antarctic glaciers remains heterogeneous [Baumhoer et al., 2018]. Studies focus either on continental scale monitoring, which usually lead to commenting the evolution of the largest glaciers of Antarctica [Walker et al., 2013, Rignot et al., 2019, Miles et al., 2022, Millan et al., 2022, Baumhoer et al., 2023] or to certain glaciers or group of glaciers that concentrate most of the attention [Baumhoer et al., 2018]."

Line 32-44: I think it would be nice to explicitly define Landfast sea-ice here and how it differs to sea-ice, some readers of The Cryosphere will not know this. Then clearly go through the mechanisms in which Landfast sea-ice can promote ice shelf advance/delay calving/stabilize ice tongues, but also how sea-ice on the open ocean can be important (e.g. buffering ocean swell) using examples from the literature. This will help set up the paper. A recent review paper on Landfast sea-ice and the references within could be a useful start here: (see [Fraser et al., 2023]; 'Antarctic Landfast Sea Ice: A Review of Its Physics, Biogeochemistry and Ecology').

We have rewritten the introduction to address this comment (L33-50).

Line 43-45: This is an excellent question.

Line 49 – "Landfast sea-ice melting" is awkward because the sea-ice can be melting but still present. Perhaps "sea-ice free conditions" or similar is better. Please revise throughout manuscript.

We corrected accordingly.

Line 58: What is wrong with radar images during this time period?

Our analysis focuses on optical imagery in open access with high spatial resolution (ASTER, Landsat, Sentinel-2) in order to be able to determine precisely the limit between the glacier ice front and landfast sea-ice. We now include some radar imagery when they can complete optical gaps.

Line 103: "ice tongue surface" – this figure does not show changes to the ice tongue surface.

We simplified the sentence: "Changes in the ice tongue front position are presented in Figure 2." (L116)

Line 187: "Figure ?"

We corrected for "Figure 5b".

Line 209: is there any bathymetry data available to confirm this? Are there any pinning points?

The presence of several islands (where the Dumont D'Urville station is located) on the western side of the glacier suggests that the bathymetry is shallower on the West compare to the East. The bathymetry of [Beaman et al., 2011] seems to confirm this, and we now cite this paper. The rift on the western side of the ice tongue always open around the same location (Figure 1 of the article) suggesting a geometrical forcing. No pinning point could be identified.

Line 216: There is little to no MYFI here.

We already answered to this question in the major comments section.

Line 223-240: There is some discussion of sea-ice trends from 1979-2016, but what about 2016-2023? There has been an exceptional decline in sea-ice, particularly this year. As a reader, I am interested what the results from this study might mean for a future with considerably less sea-ice?

We added some discussion on the recovery of the Astrolabe glacier in the future: "Considering the current decrease trend of sea ice extent in Antarctica turner2022, it is unlikely the Astrolabe would experience such an extension in the future as the recovery of the Mertz tongue is highly unlikely. The Astrolabe will most probably resume the stable cycle of yearly calving, as observe in 2000-2010 [Frezzotti and Polizzi, 2002]. It is impossible from the current knowledge on the Astrolabe glacier cycle to conclude on further retreat of the ice tongue."

Line 229-231: Please investigate the availability of imagery; I believe that there are imagery dating back to the early 1970s for the vast majority of East Antarctica that has been used in other studies.

We already answer to this question in the first comment.

Line 235: "while before 2010, 1 to 2 images per year at most are available during summer". This is not correct. MODIS allows a daily viewing of this glacier since 2000, the resolution is sufficient enough to see at least the major calving events.

We do agree that MODIS provides an exceptional dataset since 2000, but in the case of the Astrolabe glacier, it might be very difficult to distinguish between the ice tongue and sea-ice and determine reliably the ice front position because of the spatial resolution of the MODIS images. However, we attempted to monitor the ice front position (see first answer of this review) and modified this sentence to "before 2001".

MODIS link: https://worldview.earthdata.nasa.gov/?v=1524288.6381649177,-2038092.853452241,1794076.1394911779,-1918924.853452241p=antarctict=2019-03-16-T07%3A45%3A59Z

**References**

[Baumhoer et al., 2018] Baumhoer, C. A., Dietz, A. J., Dech, S., and Kuenzer, C. (2018). Remote sensing of antarctic glacier and ice-shelf front dynamics—a review. *Remote Sensing*, 10(9):1445.

[Baumhoer et al., 2023] Baumhoer, C. A., Dietz, A. J., Heidler, K., and Kuenzer, C. (2023). Icelines–a new data set of antarctic ice shelf front positions. *Scientific Data*, 10(1):138.

[Beaman et al., 2011] Beaman, R. J., O'Brien, P. E., Post, A. L., and De Santis, L. (2011). A new high-resolution bathymetry model for the terre adélie and george v continental margin, east antarctica. *Antarctic Science*, 23(1):95–103.

[Fetterer and Windnagel, 2017] Fetterer, F., K. K. W. N. M. M. S. and Windnagel, A. K. (2017). Sea ice index, version 3.

[Fraser et al., 2023] Fraser, A., Wongpan, P., Langhorne, P., Klekociuk, A., Kusahara, K., Lannuzel, D., Massom, R., Meiners, K., Swadling, K., Atwater, D., et al. (2023). Antarctic landfast sea ice: A review of its physics, biogeochemistry and ecology. *Reviews of Geophysics*, 61(2):e2022RG000770.

[Fraser et al., 2020] Fraser, A. D., Massom, R. A., Ohshima, K. I., Willmes, S., Kappes, P. J., Cartwright, J., and Porter-Smith, R. (2020). High-resolution mapping of circum-antarctic landfast sea ice distribution, 2000–2018. *Earth System Science Data*, 12(4):2987–2999.

[Frezzotti and Polizzi, 2002] Frezzotti, M. and Polizzi, M. (2002). 50 years of ice-front changes between the adélie and banzare coasts, east antarctica. *Annals of Glaciology*, 34:235–240.

[Jezek and Barry., 2013] Jezek, K. C., J. C. C. F. C. C. W. and Barry., R. G. (2013). Ramp amm-1 sar image mosaic of antarctica, version 2.

[Larour et al., 2021] Larour, E., Rignot, E., Poinelli, M., and Scheuchl, B. (2021). Physical processes controlling the rifting of larsen c ice shelf, antarctica, prior to the calving of iceberg a68. *Proceedings of the National Academy of Sciences*, 118(40).

[Miles et al., 2022] Miles, B. W., Stokes, C. R., Jamieson, S. S., Jordan, J. R., Gudmundsson, G. H., and Jenkins, A. (2022). High spatial and temporal variability in antarctic ice discharge linked to ice shelf buttressing and bed geometry. *Scientific reports*, 12(1):1–14.

[Millan et al., 2022] Millan, R., Mouginot, J., Rabatel, A., and Morlighem, M. (2022). Ice velocity and thickness of the world's glaciers. *Nature Geoscience*, 15(2):124–129.

[Rignot et al., 2019] Rignot, E., Mouginot, J., Scheuchl, B., Van Den Broeke, M., Van Wessem, M. J., and Morlighem, M. (2019). Four decades of antarctic ice sheet mass balance from 1979–2017. *Proceedings of the National Academy of Sciences*, 116(4):1095–1103.

[Walker et al., 2013] Walker, C. C., Bassis, J., Fricker, H., and Czerwinski, R. (2013). Structural and environmental controls on antarctic ice shelf rift propagation inferred from satellite monitoring. *Journal of Geophysical Research: Earth Surface*, 118(4):2354–2364.

---

## Author Comment (AC2)

Dear Editor, dear reviewers,

We thank warmly the reviewers for the careful reviews and for their comments. We propose a new version of the article taking into account the remarks of the reviewers. We explain in details the reason of our choices.

Sincerely,

Floriane Provost, on behalf of all co-authors,

NOTE: In the following document, the referee comments are in normal fonts and the answers are in blue font.

**

**Reviewer 2 comments:**

This manuscript adds to the literature on an important topic in glaciology: controls on glacier calving. Specifically, the authors construct a time-series of ice-front change and related variables (ice-flow velocity, strain rates, and sea-ice conditions) for Astrolabe Glacier in East Antarctica to better understand the causes of several calving events during the record. The general approach is useful, and the questions addressed are interesting. I think there is a fundamental issue with the analysis related to the treatment of sea ice that is important to address, and the paper could use some editing and polishing.

**Major comment: Treatment of sea ice**

The paper links the calving behaviour of Astrolabe Glacier to sea-ice forcing, which is presented fairly generally in the abstract. However, most of the rest of the paper refers to this analysis as addressing "landfast sea-ice forcing." Land-fast sea ice is a specific sea-ice configuration that is attached to land, which may mean that it provides more buttressing potential than freely floating sea ice. There are indeed several papers, cited in this manuscript for comparison, that attempt to address the role of land-fast sea ice in calving and glacier behaviour. However, this study only quantifies sea ice, not land-fast sea ice. Looking at sea-ice extent and concentration is not equivalent to assessing the presence of land-fast sea ice, and it means that the analysis cannot be as directly compared to studies that assess land-fast sea ice. Instead, the differences between the studies and the implications for differing mechanisms should be explored in more detail.

We agree that landfast sea-ice is a specific condition of sea-ice. There is no landfast sea-ice dataset that covers the period until 2021, and we hence used the [Fetterer and Windnagel, 2017] dataset of sea ice extent and concentration.

As shown now on Figure 1 (below), sea ice extent is extracted along the coast where sea ice is likely to be attached to the coast. Likewise, for sea ice concentration, the pixel is centered on the Astrolabe ice tongue and is even smaller with a size of 25 km x 25 km (Figure 1, revised), hence we do think it is fair to assume that when sea ice is present, it is connected to the land in these locations.

[Figure]

Figure 1: MODIS acquisition of January 10, 2013 centered on the Astrolabe glacier. The acquisition shows the polynya that developed in January-February 2013 at the Astrolabe glacier. The blue square represents the extent of 4,000 km$^2$ box used to extract sea-ice extent. The yellow square represents the extent of the pixel of the [Fetterer and Windnagel, 2017] product located on the Astrolabe glacier. The area of the pixel is around 600 km$^2$.

In order to validate this assumption, we used the [Fraser et al., 2020] dataset of landfast sea-ice coverage for Antarctica. This dataset is derived from MODIS imagery, two times a month, with 1 km resolution from 2000 to 2018. Overall, this dataset confirms the trend we show in Figure 5a, and b (Figure 2, below). The only exception is year 2013 where disappearing of landfast sea ice is observed in the [Fraser et al., 2020] while the extent of sea ice remains maximal in [Fetterer and Windnagel, 2017]. This difference is mostly due to difference of spatial and temporal resolution of the two dataset and to the apparition of a polynya at the Astrolabe location in early 2013 (Figure 1). Such a polynya is

[Figure]

never observed in the period 2000-2021, but for austral winter 2013.

Figure 2: Comparison between the extension of sea-ice from [Fetterer and Windnagel, 2017] and the extension of landfast sea-ice from [Fraser et al., 2020] for the two areas: a) the 4,000 km$^2$ box (in blue on Figure 1), and b) for the pixel area surrounding the glacier (in yellow on Figure 1).

I also have trouble seeing the connection between proposed physical mechanisms for ice-tongue stabilization and the analyses performed, particularly in regard to sea-ice extent. The area over which sea-ice extent is assessed is listed as being 4000 square kilometers, but the area chosen is never shown or justified. It is unclear to me how the authors determined the area over which sea-ice extent should matter to the behaviour of the ice tongue. Sea-ice concentration is taken from a pixel in a sea-ice product that covers the ice tongue, but this area is also not shown in the paper, and it is not clear whether it is centered on the ice tongue or whether all areas in the pixel are likely to affect the ice tongue. These decisions should be clearly justified and the areas shown in the paper.

We now present the extent of the 4,000 km$^2$ box and of the pixel on Figure 1 (and revised Figure 1). We choose the pixel to represent the sea-ice conditions at the glacier ice tongue location and, the 4,000 km$^2$ box to represent the sea ice conditions in a larger spatial extent which may be susceptible to buttress the Astrolabe ice tongue. Figure 2 of this response letter shows that the variations of sea ice extent [Fetterer and Windnagel, 2017] and landfast sea ice [Fraser et al., 2020] between the 4,000 km$^2$ box and the pixel is not significant in the first order.

[Figure]

**Revised Figure 1**.

Still, the fact that there is some correlation between these variables and ice-tongue behaviour is likely to be interesting. However, that correlation is not quantified. The authors claim that it is well-correlated, and there does seem to be some evidence of that in Figure 5, but it is very difficult to interpret the data from the very small panels in the figure. It would be helpful to perform the correlations, perhaps between sea-ice extent and the trend in ice-front position, for example, to better quantify the relationship.

First, we propose a new version of Figure 5 with to improve its readability, taking into account all reviewers' comments on this Figure.
Secondly, we estimated the correlation between the occurrence of calving larger than 0.25 km$^2$ (Figure 3a of this response letter) and sea ice area and concentration (Figure 3b of this response letter). The correlation is presented on Figure 3c (of this response letter). The Pearson correlation between calving event timing and sea-ice extent is 0.38 with a p-value of $1.5*10^{-10}$. The correlation is low but statically significant and confirm the trend observed between the two dataset. With a correlation coefficient of 0.13 and a p-value of $4.12*10^{-35}$, we conclude that there is a poor correlation between calving event and sea-ice concentration.

One reason for this low coefficient is likely due to the fact that calving events are "instantaneous" because iceberg detached in few days (see calving of November 2021 for example) while the free ice periods or low concentration of ice tend to last several months in the austral summer. We now integrate and discuss these values in the article.

[Figure]

Figure 3: a) Evolution of the glacier area (black dots) and timing of calving events larger than 0.25 km². b) Evolution of sea ice concentration and extent at the vicinity of the Astrolabe glacier. c) The correlation between calving event and sea ice concentration and extent.

Finally, the term "melting" appears to be used incorrectly in regard to sea ice. It seems to be used synonymously with a decrease in sea-ice concentration or extent. While this can be due to melting, these variables may also change due to sea-ice advection. Since sea-ice melting does not appear to be assessed in the study, it would be better to use a more general term.

We agree. We removed the term "melting" and replaced it by sea-ice "decrease" or "disappearing" or "sea ice free conditions" as suggesting by the other reviewer.

**Other comments:**

The manuscript is generally fairly clearly written, but there are typos and grammar issues throughout the manuscript that should be addressed. For example, hyphens between compound nouns acting as adjectives are used inconsistently, and there are many spots where verb tenses don't match the noun form. In line 8, "lead" should be "led." I am also accustomed to the term "transverse" rather than "transversal" being used for strain rates, but that may just be a convention I'm not familiar with.

We reviewed thoroughly the manuscript for typos and grammar. We corrected L8 and "transveral" for "transverse" as suggested by the reviewer.

Section 2.1.2: It would be helpful to have some indication of estimated error in the velocity correlations

Estimating the error on the velocity derived from image correlation is not an easy task. First, we propose to estimate the precision of the yearly estimation as the standard deviation of the estimated monthly velocities (presented in Figure 3a of the article). The result is presented on Figure 4a below. Secondly, the GDM-OPT-ICE service provides the displacement time series with associated RMS error on the displacement inversion [Doin et al., 2011, Bontemps et al., 2018]. The RMS error quantify how reliable is the displacement estimate. We use this to compute the velocity uncertainty as: $2 * \mu_{RMS}i^{yj}/dt$ where $\mu_{RMS}i^{yj}$ is the mean RMS error for year $yj$ and $dt$ is the delay between interpolated estimations of the displacement. The result is presented on Figure 4b below.

[Figure]

Figure 4: Estimation of the velocity precision as a) the standard deviation of the yearly velocity and b) the uncertainty of the yearly velocity from the RMS error on the displacement inversion.

The results show that both the standard deviation of the velocity and the uncertainty on the velocity decreases strongly after 2019 in the central part of the glacier, (Figure 4). One can observe that on the detaching part of the ice tongue, the standard deviation increases in 2020 and 2021 (Figure 4a). However, looking at the uncertainty on the velocity estimation (Figure 4b) this part

appear to have very low uncertainty ($< 0.1$ m.day$^{-1}$) which indicates that the estimation of the displacement is very precised in 2020 and 2021. The high standard deviation in this part of the glacier for year 2020 and 2021 is mostly due to the acceleration of the detachment.
The accuracy of the measurement is estimated from the in-situ measurement and already mentioned in the article. It is further detailed in the next comment.

Section 3.2: It would be helpful to have some more explanation in this section. I think that the GNSS measurements were averaged over the whole year to match the satellite-derived measurements, but that wouldn't be possible with the bamboo stakes, if I've understood the methods correctly. It doesn't necessarily make sense to compare a small-time slice, taken to be ground-truth, to measurements averaged over a longer period of time, but perhaps that's what the comments on lack of seasonal variation are trying to address. It's also not always reasonable to compare point measurements to those averaged over a large spatial area. Finally, I can't quite figure out what the last two sentences in this section are trying to say. I suspect all the analyses discussed in this section are reasonable, but I can't quite figure that out based on what is written.

We propose to detail this analysis in the supplementary information of the manuscript, with the following explanations. The GNSS campaigns available for this study are years 2018 (points 1 to 8, Figure 5a) and 2021 (points 9 to 12, Figure 5a). There is a gap of measurement in 2019 and 2020 due to the COVID crisis. Figure 5b presents the annual time series of the GNSS velocity. The first observation is that the time series do not exhibit particular seasonal variations in this part of the glacier (Figure 5b) for these two years. The second observation is that the velocity seems to be constant over time. Indeed, points 5 and 9 are located in the same area and exhibit a velocity of $1.27 \pm 0.14$ m.day$^{-1}$ in 2018 and $1.28 \pm 0.11$ m.day$^{-1}$ in 2021 (Figure 5b). Similarly, points 8 and 12 exhibit the same range of velocity, as well as points 7 and 10 between the two campaigns (Figure 5b).

The second set of in-situ measurements are 16 bambou sticks installed for one week between January 31, 2020 and February 7, 2020. The velocity of the bambou sticks is derived from the measured positions at the beginning and end of the campaign. Although this campaign is relatively short in time, it provides an interesting profile from the glacier limits (point A, Figure 6b) to the center (point A', Figure 6a). As the GNSS time series do not exhibit seasonal variations nor major variation from 2018 to 2020 (considering neighboring) points, we assume the velocity measured by the bambou sticks is representative of the glacier velocity. To confirm this assumption, we compare the bambou sticks velocity to the GNSS velocity of 2018 and 2021 projecting the GNSS position along the bambou profile. The results are plotted in Figure 6b, and we observe that the GNSS velocity are in agreement with the bambou stick velocity (Figure 6b). To explore further the variation of the velocity through time we also extracted the velocity from the MEaSUREs dataset available from 2000 to 2018

[Figure]

Figure 5: a. Location of GNSS permanent stations: 1 to 8, stations installed in 2018 and 9 to 12, stations installed in 2021. For each measurement point, one year of date is available in 2018 or in 2021. The arrows show the total displacement of the point measured with the GNSS stations (blue) and with image correlation (blue) for years 2018 and 2021. b. Evolution of the velocity for the year of acquisition (2018 or 2021) for each of the 12 GNSS station.

[Gardner et al., 2018]. Except for years 2000, 2006 and 2012, the two datasets are in very good agreement (Figure 6c).

Finally, we compare the velocity derived from the GDM-OPT-ICE dataset and the bambou sticks (Figure 6d). The RMS error between GNSS measurement and 2017 and 2018 GDM-OPT-ICE estimation is 0.76 m.day-1. The GDM-OPT-ICE estimation is particularly poor on the edge of the glacier, while it significantly improves toward the center (Figure 6d). From year 2018, the GDM-OPT-ICE results slightly improve toward the center of the glacier tongue (Figure 6d). From 2019, the accuracy of the GDM-OPT-ICE improves (RMS < 0.25 m.day$^{-1}$) and one can observe that the two datasets are in good agreement. In 2020 and 2021, the GDM-OPT-ICE velocities tend to be in agreement with the bambou velocity only on the edge of the glacier (Figure 6d, toward point A). In the center of the glacier tongue, GDM-OPT-ICE velocity have larger magnitudes (1.25-1.5 m.day$^{-1}$) than the bambou stick velocity (1.20-1.25 m.day$^{-1}$). However, comparing the early months of 2020 (January to March 2020), the derived velocity is in agreement with the bambou sticks velocity measured during this period.

Lines 158-169 says: "It can be noted that compressional strain rates are measured from 2017 to 2020 at the terminus of the glacier tongue with strain rate

[Figure]

Figure 6: Comparison between in-situ data (i.e. GNSS measurement of 2018 and 2021 and bambou sticks campaign of 2020) with the location of the measurement points in (a) and comparison of the derived mean velocity in (b). The bambou measurements are compared with MEaSUREs yearly velocity [Gardner et al., 2018] from 2000 to 2018 (c) and to the yearly estimation of the velocity from the GDM-OPT-ICE products from 2017 to 2021 (d).

larger than $0.001$ day$^{-1}$ while it is not observed anymore in 2021." Strain rates are usually positive in extension, so compressional strain rates could not, by definition, be larger than 0. Is this an absolute value, or is there a different convention being used here?

This is an absolute value as we refer to "compressional strain rate" we then mentioned the absolute value of the strain rate.

Figures:

There are several spots where figures are not referenced correctly. E.g. in section 2.1.1, it seems like the references to Fig. 1 should refer to Fig. 2, and in section 3.4, Fig. ?? should be corrected to the figure number.

We apologize for this and corrected accordingly.

Figure 2: It would be helpful if the y-axes were the same in panels e-g to facilitate comparison between panels.

We corrected the figure accordingly.

Figure 4: I don't find Fig. 4a very helpful, because the panels are too small to clearly see what is discussed in the text. Consider showing just a few panels that are necessary to the analysis, and shifting the rest of the panels in larger form to supplementary information. I think Fig. 4b is a very clever way to display the information.

We prefer to keep the figure 4a as it is, we do think the information is readable. We modified figure 4c in order to point out the fissures appearing in June 2021.

In general, it would be helpful to have more labels in the figures that correspond to what is discussed in the text. For example, the discussion talks about the "main rift" and refers us to Fig. 3 on line 194. I can make some guesses based on Figure 3 about what the main rift is, but I would rather have a label or two that helps me know exactly what the authors are referring to. It would be helpful to have those labels in Figure 4, as well.

Labels/boxes are already present on Figure 3c corresponding to "A: the main rift" and "B: secondary rift". However, we tried to improve this on all figures. We added those label on all sub-figures when the rifts are visible on Figure 3, we added the location of the rift on the revised version of Figure 1. We also indicated the network of fissures in Fig. 4c. We hope this help the reader.

**References**

[Bontemps et al., 2018] Bontemps, N., Lacroix, P., and Doin, M.-P. (2018). Inversion of deformation fields time-series from optical images, and application to the long term kinematics of slow-moving landslides in peru. *Remote Sensing of Environment*, 210:144 – 158.

[Doin et al., 2011] Doin, M.-P., Guillaso, S., Jolivet, R., Lasserre, C., Lodge, F., Ducret, G., and Grandin, R. (2011). Presentation of the small baseline nsbas processing chain on a case example: the etna deformation monitoring from 2003 to 2010 using envisat data. In *Proceedings of the Fringe symposium*, pages 3434–3437. ESA SP-697, Frascati, Italy.

[Fetterer and Windnagel, 2017] Fetterer, F., K. K. W. N. M. M. S. and Windnagel, A. K. (2017). Sea ice index, version 3.

[Fraser et al., 2020] Fraser, A. D., Massom, R. A., Ohshima, K. I., Willmes, S., Kappes, P. J., Cartwright, J., and Porter-Smith, R. (2020). High-resolution mapping of circum-antarctic landfast sea ice distribution, 2000–2018. *Earth System Science Data*, 12(4):2987–2999.

[Gardner et al., 2018] Gardner, A. S., Moholdt, G., Scambos, T., Fahnstock, M., Ligtenberg, S., Van Den Broeke, M., and Nilsson, J. (2018). Increased west antarctic and unchanged east antarctic ice discharge over the last 7 years. *The Cryosphere*, 12(2):521–547.

---

## Author Response (AR2)

Dear Reviewer,

We would like to thank you for your review and pertinent suggestions to improve the manuscript. We have taken into account all of them in the new version of the manuscript. We have also performed an important review of the grammar and spelling in order to correct the remaining mistakes and improve the readability of the article.

We answer below to all the points raised by the reviewer in blue.

Sincerely,
Floriane Provost, on behalf of all co-authors.

**

Thanks to the authors for a thorough response to my revision suggestions; I think the manuscript has improved a lot. I've made a few more suggestions below, including a couple of incorrect statements that will need to be corrected before publication, as well as a few more suggestions to improve readability.

Use of sea ice vs. fast ice: To be honest, I'm still not completely convinced by the arguments in the review response about why the sea-ice patterns can be regarded as the same as fast-ice patterns. There are only a few years available for correlation and they're not all strong. That being said, I think the modifications to the text take care of this issue; it's fair to refer to these as being sea-ice changes. It might be helpful to include some of the info from the review response in the discussion suggesting that fast-ice patterns are similar to the measured sea-ice patterns, but that future work would benefit from better fast-ice data availability.

There is 18 years of overlap between the sea-ice extent dataset from Fetterer et al., 2017 and the fast-ice dataset from Fraser et al., 2020. Over 12 years (from 2000 to 2012), the general trend of seasonal disappearing of sea-ice and fast-ice during the summer season is observed on both datasets. From 2013 to 2018, the two dataset are in agreement that sea-ice and fast-ice is not disappearing except for austral summer 2012-2013 and summer 2015-2016. For austral summer 2015-2016, the disappearing of sea-ice and fast-ice at the vicinity of the Astrolabe glacier is clearly observed on Landsat images. For austral summer 2012-2013, we already discussed this in the previous review response, showing the development of a polynya around the Astrolabe glacier which may explain the difference between the two datasets due to their different spatial and temporal resolution. This is the only difference observed in the 18 years of comparison if we only account for the cycle of presence/absence of sea-ice/fast-ice at the vicinity of the Astrolabe. This cycle is also the only criteria that is

compared with the Astrolabe calving cycle in the paper. We do not consider any other properties (i.e. thickness, regional extent, etc.).

Finally, we do agree that a deeper analysis is needed but this is beyond the scope of the present work. We do think the current version of the manuscript presents a fair comparison between sea-ice changes and the Astrolabe tongue calving cycle. It is also in line with recent studies that made use of the same dataset to analyze the calving cycle of other ice tongues (Gomez-Fell et al., 2022). We do think that without any detailed analysis on the local fast-ice/ sea-ice evolution, comparing the two datasets may lead to more confusion rather than clarification. We prefer to keep the analysis as such and point that a deeper investigation including the role of fast-ice, especially in the rifts, is needed.

Use of airborne imagery: I think this is a good addition to the paper, and that the other reviewer was correct that using the full span of available historical data is required for some of the claims in the paper. However, I don't think the use of this has been adequately explained. First, any statement that lists 1947 as the first date must acknowledge the use of airborne imagery (e.g. Lines 4-5: Add "…airborne and satellite imagery," as the 1947 imagery is not from satellites. Also true in line 54).

We agree, and corrected them accordingly.

Second, the use of these data needs a bit more info in the methods. Most available historical air photos are not currently accurately georeferenced, which makes it more challenging to delineate terminus positions. The methods just mention using the "sketch of the photography," and just the resulting ice front is shown in the figure. More info about how that sketch was made and how the georeferencing was done should be included.

We attempt to clarify the sources and the methodology adopted for this image (L 68-71):
« In 1947, the US Navy Operation Highjump took several aerial photographs of the Adélie coast, including over the Astrolabe glacier. We used a sketch derived from the photograph (https://archives-polaires.fr/idurl/1/14865) to extract the ice front position (Figure 1c). We manually coregister the photograph and attempt to compensate for most of the distortions, although significant shifts remain visible. We hence account for ± 1 km of error in the ice front position. »
The uncertainty on the terminus position is significant for this image but we are certain it can support the statement that the tongue terminus position reaches an unprecedentedly observed position from 2019 to November 2021.

Grammar: There are still quite a few issues with grammar in many spots in the paper. In Line 14, the hyphen was correct before the edit: "sea-level rise" should be hyphenated, as should

all pairs of nouns acting as compound adjectives before another noun. Other examples include "ice-tongue fracturing" and "sea-ice extent" in the abstract and elsewhere. "Sea-ice-free" should have two hyphens. Many examples are hyphenated correctly in the document, but it is not consistent. Note that hyphens aren't necessary when the two nouns are not modifying another noun. For example, "sea ice" used on its own does not need to be hyphenated, but it does when it modifies "extent." While not always necessary, I think this would improve the readability of the paper.

We thoroughly reviewed all hyphens and corrected them accordingly. We also reviewed thoroughly the entire text and corrected many sentences in order to improve the readability of the paper.

Some other minor grammar points:

Lines 20-24: Format citations correctly and revise grammar; does not currently make sense

We corrected the format of the citations and revised the grammar.

Line 44: Adjust to "i.e. sea ice fastened to the glacier/to the coastline"

We corrected them accordingly.

Line 51: Should be a period rather than a question mark

We corrected them accordingly.

Line 133: "velocity" is singular; should use "velocity is" rather than "velocity are," or conversely use "velocities are"

We corrected for « velocities are ».

Line 260: "traducing" is not the correct word here

We changed « traducing » to « reflecting ».

 Lines 268-269: Put citations in parentheses. Note that I have not marked everything that should be corrected.

We corrected accordingly and reviewed and corrected the format of all citations including in the "References" section.

Message: I really like the last sentence of the introduction, but those ideas don't come through quite as clearly in the abstract. Consider revising the abstract, perhaps by using this sentence.

We rewrote the abstract to include the statement at the end of the introduction:

« The recent calving of the Astrolabe glacier (Terre Adélie, East Antarctica) in November 2021 is an opportunity to better understand the processes leading to ice tongue fracturing. The archive of Sentinel-2 optical images is used to measure the ice motion and the ice strain rates for the period 2017-2021 in order to document fractures and rift evolution that lead to the calving. Additionally, the long-term evolution of the Astrolabe ice tongue is mapped with airborne and satellite imagery from 1947 to November 2021. These observations are compared with sea-ice extent and concentration measurements. We show that calving occur almost systematically at the onset or during the melting season and that a significant change in the sea ice periodicity surrounding the Astrolabe glacier in the last decade (2011-2021) in comparison to the previous observations (1979-2011) lead to a change in the Astrolabe calving cycle. Indeed, one can observe a decrease of the duration of sea-ice-free conditions during austral summer after 2011 at the vicinity of the glacier that seems to have favored the ice tongue spatial extension. However, the analysis of strain rate time series revealed that the calving of November 2021 (20 km$^2$) occurred at the onset of sea ice melting season but resulted from the glacier dislocation that took place suddenly in June 2021 in the middle of the winter. **These observations suggest that although sea ice acts as a protection favoring spatial extension of glacier ice tongue, its buttressing is not sufficient to inhibit rifting and ice fracturing**. »

Other comments:

Lines 67-68: I believe Landsat 4 ceased transmissions in 1993, and therefore could not be the source of a 1999 image

The two cloud-free images of 1999 are from Landsat-7 archive. We corrected the text.

Line 72: Landsat-8 did not launch until 2013

Indeed, we actually did only mention « Landsat » in the text. We now precise that it is Landsat-7 and ASTER for the period 2000-2013 (L 73).

Figure 2a: I realize that I can reference Figure 1 to see the glacier outline, but it would be useful to have that mapped here.

We do not think this is necessary. We believe the reader can easily refer to Figure 1.

Line 125: Reference Figure 1c here for the profile positions

We refer here to the location of the profiles AA', BB', and CC' that are presented on Figure 1b. We add a sentence to refer to the terminus position: « The delineation of the terminus positions are mapped on Figures 1c-f. »  We hope this is much clearer for the reader.

145-146: "in 2017 and 2018, the limit between stable ice and the flowing ice tongue is retrieved in the wrong position with the GDM-OPT-ICE products." I think you're saying that the GDM-OPT-ICE results show velocities in the wrong direction near the western shear margin. That looks to be true in all years, not just 2017 and 2018. It would help to clarify this.

Here we do not refer to the western shear margin but to the 9 bamboo stick locations at the western side of the bamboo stick profile. We try to precise the comment:

« One can observe that the gradient of velocity from the western border to the center of the glacier is well retrieved with the GDM-OPT-ICE products of 2019-2021 (Figure 3b) while in 2017 and 2018, the limit between stable ice and the flowing ice tongue is retrieved in the wrong position with the GDM-OPT-ICE products. Indeed, in 2017, the GDM-OPT-ICE velocity of the 9 bamboo sticks located on the western side of the profile is almost null for all locations. Conversely, a progressive increase of the velocity is measured during the bamboo sick campaign (Figure 2b) and by the ITS_LIVE products (Figure 2c). The same is observed in 2018, although the velocity derived from GDM-OPT-ICE is slightly larger than in 2017 (Figure 3b). The small number of cloudless Sentinel-2 acquisitions for those years may explain the low RMS error of these two years, as well as the wrong estimation of the ice tongue limit. »

For years 2019-2021, the gradient of velocity from the western margin to the center of the glacier is accurately retrieved.

Edit the caption of figure 5. I think the second sentence is explaining panel b but is not labeled that way, and the arrows showing the main rift and secondary fractures are shown in c and d, not b and c

We corrected the caption to take into account the comment of the reviewer.

Line 176: Add "with absolute value of the strain rate" larger than… for clarity

We corrected them accordingly.

Figure 6 caption: The 25 km x 25 km pixel is in a dotted yellow line in Figure 1, not blue

We corrected them accordingly.

Lines 259-261: "…lead to severe modifications of the sea ice production and location, traducing regional changes in the oceanic and atmospheric currents (Campagne et al., 2015). This event is likely at the origin of the transition of the sea ice seasonal cycle at the Astrolabe glacier." It would be helpful if this section was more specific, for the benefit of a reader not deeply familiar with the Mertz ice tongue calving. Can you explain a bit more about what the modifications were, and why this is likely the origin of the transition of the sea ice cycle?

We added subsequent details on the Mertz calving and its impact on its western region (L310-317). We hope this answers the reviewer's comment and helps the reader to understand the regional context. As stated at the end of the article, the impact of the Mertz glacier calving along the Adélie coast remains only partially known as most studies focused so far on the Mertz glacier surroundings and/or on the one or two years following the calving events.